# Circular wood use can accelerate global decarbonisation but requires cross-sectoral coordination

Eilidh J. Forster [1] ✉, John R. Healey [1], Gary Newman[2] & David Styles[1,3]

Predominantly linear use of wood curtails the potential climate-change mitigation contribution of forestry value-chains. Using lifecycle assessment, we show that more cascading and especially circular uses of wood can provide immediate and sustained mitigation by reducing demand for virgin wood, which increases forest carbon sequestration and storage, and benefits from substitution for fossil-fuel derived products, reducing net greenhouse gas emissions. By United Kingdom example, the circular approach of recycling medium-density fibreboard delivers 75% more cumulative climate-change mitigation by 2050, compared with business-as-usual. Early mitigation achieved by circular and cascading wood use complements lagged mitigation achieved by afforestation; and in combination these measures could cumulatively mitigate 258.8 million tonnes $CO_2$e by 2050. Despite the clear benefits of implementing circular economy principles, we identify many functional barriers impeding the structural reorganisation needed for such complex system change, and propose enablers to transform the forestry value-chain into an effective societal change system and lead to coherent action.

The forestry value chain is a key pillar of the 'circular economy' (CE)[1–5] as a major source of renewable biomaterial and can deliver multi-faceted climate-change mitigation benefits, including carbon sequestration and avoided emissions from fossil-fuel-derived product substitution[6]. Global consumption of primary processed wood products is predicted to rise by between 60%[7] and 170%[8] by 2050, but current value-chains are suboptimal[9] and would not sustainably meet future demand under the predominant linear economy model[10]. There is considerable scope to increase the sustainability of forestry value chains and increase their contribution to achieving net-zero greenhouse gas (GHG) emissions[11], in alignment with Paris Agreement[12] goals. Currently, decarbonisation and circular economy policies tend to have a narrow sectoral focus, e.g. on the development of zero-emissions energy generation, afforestation for residual carbon offsetting, or increasing use of renewable materials in place of fossil-fuel-derived materials[13]. Yet there appears to be little cross-sectoral

integration of these sustainability objectives and little focus on more circular use and recycling of wood as part of a coherent decarbonisation strategy[14,15]. There is, therefore, a need for prospective lifecycle assessment (LCA) with widely-defined (multi-use) boundaries, like the present study, to quantify the additional mitigation potential of implementing CE principles in the forestry value-chain, to provide critical evidence for systemic change necessary to deliver rapid and sustained climate-change mitigation.

Transitioning to a CE is a 'wicked' problem[16–18] requiring large socio-economic structural changes and industrial re-organisation[19]. Hundreds of organisations operate in the forestry value chain at a national level; increasing to many thousands at a global level. To overcome the 'organising' challenge that results in suboptimal climate-change mitigation, the forestry value-chain must function as a societal change system (SCS), with shared overarching goals and principles guiding coherent and convergent action[20]. A high-functioning SCS

[1]School of Environmental and Natural Sciences, Bangor University, Bangor, Gwynedd LL57 2UW, UK. [2]Woodknowledge Wales Ltd., Ffarm Moelyci, Felin Hen Road, Tregarth, Gwynedd LL57 4BB, UK. [3]School of Biological & Chemical Sciences and Ryan Institute, University of Galway, Galway H91 TK33, Ireland. ✉e-mail: afp239@bangor.ac.uk

needs to perform seven critical change functions: system visioning; system organising; resourcing; learning; measuring; advocating; and prototyping to achieve change effectively[20]. However, no analysis of the forestry value chain as a SCS has yet been performed. Barriers to CE have previously been identified and catalogued as external factors impacting action at an organisation level[21,22] (e.g., political, economic, sociological, technological, legal and environmental factors), but not against value-chain system-functioning criteria. SCS analysis of the forestry value chain is needed to identify attributes that limit the system-change functions to determine pragmatic steps to overcome these barriers.

This study aims to address these two important evidence gaps, to inform effective policy and industry actions targeting net-zero GHG emissions. First, we identify wood-use strategies that substantially increase climate-change mitigation by applying dynamic, consequential LCA[23] to four wood-use scenarios over a 28-year study period to 2050. Second, we propose key enablers of system change by interviewing forestry value-chain actors on perceived barriers to circularity and by analysing responses against a societal change matrix (of functions needed to achieve system change[20]). By combining insights from LCA and SCS analysis, we identify what to change and how to change it.

## Results and discussion
### Analysis of wood use strategies in a UK context
Differentiating value chains using LCA and SCS requires high-resolution data—achieved here using a detailed case study of the UK domestic forestry value chain. UK softwood production and processing supplies around 20% of domestic needs; the UK relies heavily on imports but exports very little[24]. Since the wood flows out of UK forests mostly remain within national boundaries, traceability is high—making the UK an ideal case study for LCA of a whole forestry value-chain (from the forest through to harvested-wood-product (HWP) end-of-life). We use consequential LCA[23] to assess the climate-change mitigation impact—measured as 100-yr global warming potential (GWP) expressed as carbon dioxide equivalent ($CO_2e$) emissions—for business-as-usual ('BAU') UK-forestry value-chain softwood use during 2022–2050, accounting for the effects of progressive industrial decarbonisation (i.e. increasing deployment of zero-emissions technology). We then assess three alternative wood-use scenarios to calculate the climate-change mitigation impact of enhanced 'cascading', 'circular' and 'cascading&circular' uses (see below, Methods). Enhanced 'cascading' involves more production of sawn wood and less production of wood panels from virgin wood in the UK. Enhanced 'circular' involves the manufacture of recycled medium-density fibreboard (MDF) from recovered waste MDF (using currently available commercial technology). 'Cascading&circular' combines the two. Confining scenarios to proven cascading and circular uses ensures robust LCA and is a conservative approach (alternative options discussed in Supplementary Methods 1). We modelled the impact of delaying the implementation of these scenarios from year 5 to year 10.

### Carbon emissions distribution across value-chain
To observe relative GWP contributions of different components in the value-chain for the 'BAU', 'cascading', 'circular' and 'cascading&circular' scenarios, we analyse relevant $CO_2e$ emission sources (Scopes 1–4, defined in Fig. 1) in the year 2035.

In all four scenarios, wood panel production is the biggest $CO_2e$ emitter—contributing 63% of 'BAU' net Scope 1–3 GWP burden, of which around 40% is attributed to Scope 3 emissions from the use of resins (Fig. 1). 'BAU' MDF production also involves high Scope 1–3 emissions and high consumption of virgin material (see below, Methods), implying substantial opportunity to reduce GWP burden. 'Avoided emissions—product substitution' contributes the largest GWP credits (i.e. emission reductions) across all scenarios (delivered by

woodfuel substituting for fossil fuel and sawnwood substituting for concrete in construction, in similar magnitudes). Avoided emissions offsets all value-chain emissions and delivers net negative Scope 1–4 emissions of −2.1 to −3.9 million tonnes of $CO_2e$ in 2035.

Compared to 'BAU', the 'cascading' wood flow scenario has higher 'sawmill' emissions, more HWP carbon storage, and lower 'wood panel production' emissions, resulting in 35% lower Scope 1–3 emissions overall (Fig. 1). Therefore, greater cascading use is beneficial from a national emissions accounting perspective (since these Scope 1–3 emissions are UK-attributed). However, it does not significantly enhance net GWP reduction (i.e. Scope 1–4 emissions) since imported HWP volumes (and associated emissions) adjust to balance changes in UK production and maintain stable UK consumption (Fig.1). The 7% additional GWP reduction in the 'cascading' scenario derives from increased HWP carbon storage in UK-produced sawn wood. 'Avoided emissions—product substitution,' credits are unchanged from 'BAU' to 'cascading' since UK-HWP consumption is static and emissions from UK- and imported-HWP production are equivalent.

### Circular use reduces the demand for virgin wood
The greatest differentiating factor across the four scenarios is the net carbon sequestration gain from reduced harvesting (in non-domestic forests) in the circular wood flow scenarios ('circular' and 'cascading&circular') (Fig. 1). Circular wood flow scenarios reduce consumption of virgin wood relative to 'BAU', increasing 'avoided emissions—reduced harvest' and delivering GWP credits that offset all (125% of) 'BAU' Scope 1–3 emissions. Circular scope 1–3 process emissions are also reduced (because of lower energy demand for recycled-MDF production compared to MDF production), along with imported HWP emissions (due to a net increase in UK-HWP production). Despite reduced avoidance of fossil-fuel emissions due to less waste-wood fuel availability than under 'BAU', the 'circular' and 'cascading&circular' scenarios achieve 85% and 87% larger net (Scope 1–4) GWP reductions in 2035 than under 'BAU', and 73% and 75% larger reductions than the 'cascading' scenario, respectively.

The largest GWP reduction is achieved by the combined 'cascading&circular' scenario, which is 1% more effective at reducing GWP impact than 'circular' alone (Fig. 1). This subtle enhancement is because the 'circular' scenario also achieves improved cascading use compared to 'BAU' wood use due to redirection of virgin material at the forest gate from wood panel production to sawmills. Therefore, the additional cascading material flow enhancements in the combined 'cascading&circular' scenario only led to marginal further GWP reductions. Overall, all modelled cascading and circular changes to material flow from 'BAU' result in larger GWP reductions, both when considering net Scope 1–3 and net Scope 1–4 GWP emissions.

### Climate-change mitigation of circular wood use is resilient to industrial decarbonisation
The relative GWP performance across the four wood-use scenarios in 2035 (Fig. 1) carries over to the dynamic annual net GWP impacts throughout the period 2022–2050 (Fig. 2a–d). Every year, the 'cascading&circular' scenario delivers the smallest GWP burden (or largest GWP reduction), followed closely by 'circular', whereas 'cascading' provides only marginal additional GWP-reduction over 'BAU'. We apply the same progressive industrial decarbonisation factors to the relevant value-chain components (i.e. processing emissions) in all four scenarios so that both net Scope 1–3 GWP burdens (Fig. 2a, b) and net Scope 1–4 GWP reductions (Fig. 2c, d) shrink over time—the latter reflecting diminishing 'avoided emissions—product substitution'. Net Scope 1–4 GWP reductions for 'circular' and 'cascading&circular' wood-use scenarios are less affected by industrial decarbonisation since diminishing (avoided emission) factors do not apply to the dominant biogenic carbon storage credits ('change in HWP C storage' and 'avoided emissions—reduced harvest').

 

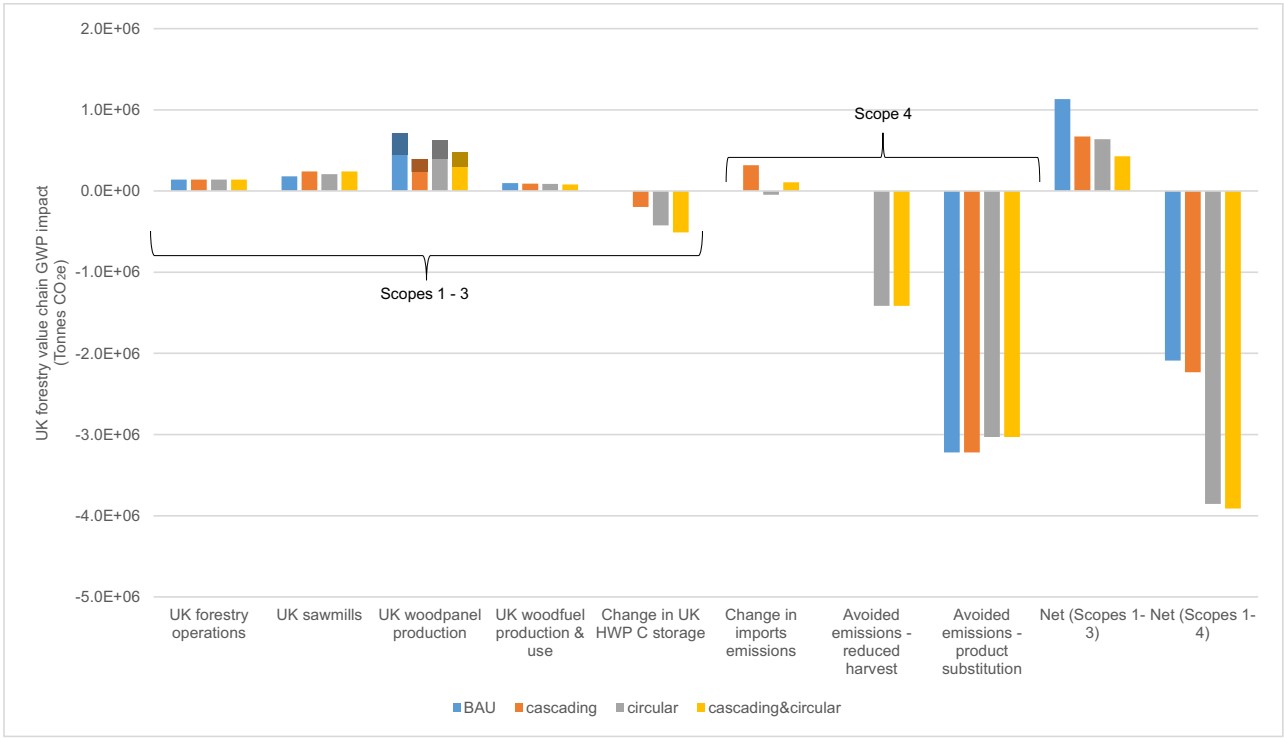

**Fig. 1 | Distribution of global warming potential (GWP) impacts for four UK forestry value-chain scenarios in the year 2035.** Bars represent GWP impact in tonnes of carbon dioxide equivalent emissions ($CO_2e$) for four modelled UK wood material flow scenarios (Methods, Fig. 5) under decadal decarbonisation assumptions (Supplementary Information). 'BAU' represents the 2022 UK wood material flow. Enhanced 'cascading' involves higher UK sawnwood production and lower UK wood panel production. Enhanced 'circular' involves the manufacture of recycled medium-density fibreboard (MDF) from recovered waste MDF. 'Cascading&circular' combines the two. Emissions sources are separated into value-chain components on the x-axis (and represented in the LCA boundary diagram, Fig. 6). 'UK Forestry operations' includes tree nurseries, site preparation, tree planting, harvesting and timber transport from forest to wood processor. 'Change in UK HWP C storage' is the net gain or loss of carbon (C) stored in HWP manufactured in the UK from UK-produced timber. 'Change in import emissions' is the net gain or reduction of transport emissions associated with a change in volume of imported HWP. 'Avoided emissions−reduced harvest' is increased carbon storage in forests as a result of lower demand for virgin material leading to lower harvest rates. 'Avoided emissions−product substitution' is emissions from concrete production that are avoided due to sawn wood substituting for concrete in construction, and emissions from burning fossil fuels that are avoided due to woodfuel use. Emissions are grouped under 'Scopes'[48,49]. Scope 1 is direct process emissions. Scope 2 is emissions from the generation of energy imported into processes. Scope 3 is emissions from manufacturing of materials imported into processes (dominated here by resins used in wood panel production−shown with shading in wood panel production bars; scope 3 emissions from other processes are too small to observe). Scope 4 is avoided emissions, in this case from product substitution, reduced harvest, and changes in imports. Scope 4 is typically not included in industry decarbonisation target-setting. However, it is important for quantifying the consequential whole-lifecycle impact of system changes, hence net GWP impact is presented, including and excluding Scope 4 emissions. Source data are provided as a Source Data file.

## Early implementation optimises impact

Since annual Scope 1−4 GWP reductions shrink over time, prompt transition to 'circular' and 'cascading' wood use in the first five years achieves both an earlier and faster rate of cumulative GWP reduction than delaying action by a further five years (Fig. 2c, d). When implemented in year 5, the 'cascading&circular' scenario achieves an average annual GWP reduction of −3.7 million tonnes $CO_2e$ per year post-implementation and a cumulative reduction of −96.6 million tonnes $CO_2e$ by 2050 (Figs. 2c and 3c). However, when implemented in year 10, the respective GWP reductions are −3.4 and −87.5 million tonnes $CO_2e$ (Figs. 2d and 3d).

## Circularity creates a carbon sink

A 'net-zero' (Scopes 1−3) GWP forestry value chain is only achievable by 2050 if circular wood use is implemented. Dynamic results show that annual Scope 1−3 GWP impact (typically the basis for industry-level decarbonisation targets) reduces over time with industrial decarbonisation (Fig. 2a, b). However, these emissions will only reach or surpass net zero by 2050 if 'circular' or 'circular&cascading' wood use is implemented in parallel with industrial decarbonisation. Implementing the 'circular' or 'circular&cascading' scenarios achieves (Scope 1−3) net zero by 2050 and thereafter becomes a net carbon sink (Fig. 2a, b).

'BAU' annual net Scope 1−3 GWP impact will eventually become net zero when the industry fully decarbonises. 'BAU' Scope 1−4 GWP impact will become net zero at the same time since imported-HWP countries are assumed to decarbonise at the same rate as the UK, so 'avoided emissions−product substitution' will also become zero. However, while Scope 1−3 emissions and 'avoided emissions−product substitution' diminish over time, circular wood use continues to provide annual GWP credits via 'HWP C storage' and 'avoided emissions− reduced harvesting'. Therefore, only 'circular' or 'cascading&circular' wood use can lead to the forestry value-chain becoming an enduring (Scope 1−4) net carbon sink (even before considering the potential contributions of afforestation and bioenergy with carbon capture and storage (BECCS)).

## Circular wood use complements afforestation as a 'net zero' strategy

Implementing 'circular' or 'circular&cascading' wood use achieves considerable immediate GWP reduction, followed by a smaller yet sustained reduction to 2050 (Figs. 2c, d and 3c, d). In comparison, the GWP reduction attainable through 'afforestation' (defined in Fig. 3) builds gradually and increases pace as 2050 approaches (Fig. 3c, d). The best-case combined GWP impact of 'afforestation' and

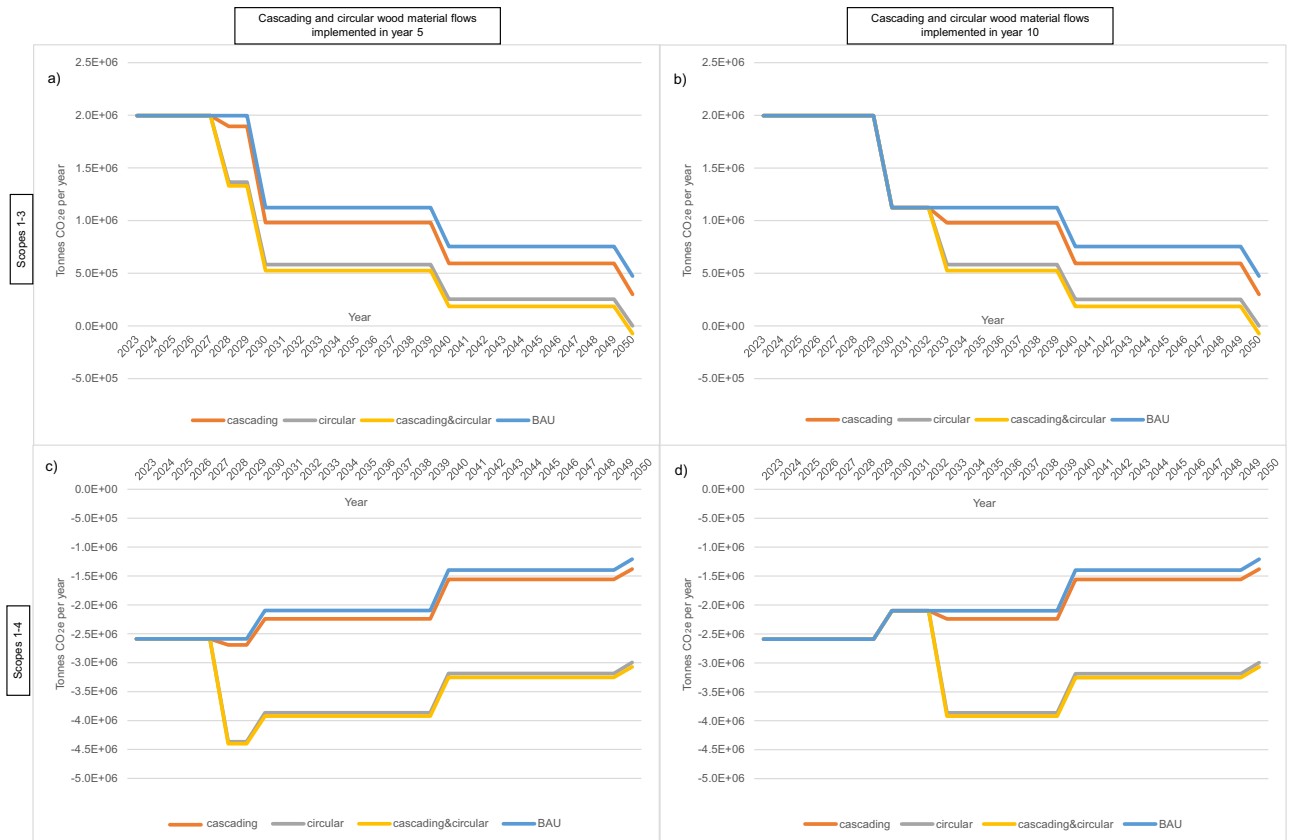

**Fig. 2 | Annual global warming potential impact of alternative UK forestry value-chain scenarios.** Global warming potential (GWP) impact is based on a static wood harvest of 9.5 million green tonnes per annum in the UK over the study period. Stepwise, decadal decarbonisation is applied in 2030, 2040 and 2050, which causes the stepped shape in the graphs. Graphs allow comparison of net GWP impacts, without (**a**, **b**) and with (**c**, **d**) Scope 4 emissions are included to observe the important and changing contribution of Scope 4 (avoided) emissions over time as linked industries decarbonise. The graphs also enable comparison of net GWP impacts of implementing circular and cascading scenarios after 5 years (**a**, **c**) and 10 years (**b**, **d**) to observe the impact of delaying action. Source data are provided as a Source Data file.

'circular&cascading' wood use is −258.8 million tonnes $CO_2e$ by 2050, while 'afforestation' alone will only achieve −162.3 million tonnes $CO_2e$ by this year (Fig. 3c). Significant further GWP reduction from 'afforestation' will continue to accrue after 2050 from ongoing carbon sequestration in forest growth, and later from HWP[6]. Therefore, as part of a national net-zero strategy, circular wood use is complementary to the GWP impact of afforestation.

## Barriers to circularity

We gathered information on experiential knowledge of barriers to decarbonisation and transitioning to a CE via in-depth semi-structured interviews with seventeen individuals from diverse organisations across the UK forestry value-chain (tree nursery, tree planting, forest management, harvesting, sawmilling, wood panel manufacturing, biotechnology, carbon markets, land agents and trade organisations). We organised and defined the barriers identified by participants under the seven change-function categories needed for an effective SCS[20]. Twenty-four barriers to change are identified and indicate weaknesses in the performance of every SCS change function in the forestry value chain (Tables 1 and 2).

Shared 'system visioning' is the bond needed to create coherent action in multi-stakeholder collaborations[25], beginning with a broad global vision that provides common guidance for principles adapted to local conditions[20]. During interviews, we found there is no clear unifying global vision for the role of forestry in a net-zero CE. Rather, a narrow focus on the fragmented implementation of zero emissions technologies to decarbonise particular operations and subsectors predominates. Participants reported organisation strategy focussing only on decarbonisation or no strategy at all (Table 1). Thus, despite being long-established, we deduce that the UK forestry value chain is not organised appropriately to facilitate complex system change. A SCS requires 'organising' of effort and stakeholders in ways that provide coherent aggregation of voice at scale in order to be heard. This can include collaborations and networks of organisations[20], such as trade organisations, which are numerous in the UK forestry value chain. During interviews, participants reported a lack of willingness to collaborate and co-ordinate between value-chain stakeholders and stakeholder groups (Table 1), which limits the organising of efforts, shared learning, and therefore effective SCS function.

Mind-set and capacity for learning are key at the individual, organisational, and system levels. Addressing complex change challenges demands new ways of thinking about problems and of taking action. 'Learning' change initiatives can include advancement and sharing of knowledge from the prototyping of new technologies or business models; they can also include multi-stakeholder networks that focus on establishing events, interactions, and publications to support realising the vision[20]. During interviews, we identified a culture where sharing of knowledge and experience is not consistent across the value chain. Interviewees reported slow public release of new forestry research; and a number of important knowledge gaps, such as the exploitable material properties of alternative commercial tree species and the carbon impact of silvicultural practises across different soil types. Wood processors conveyed a negative attitude towards shared learning on the implementation of decarbonisation initiatives

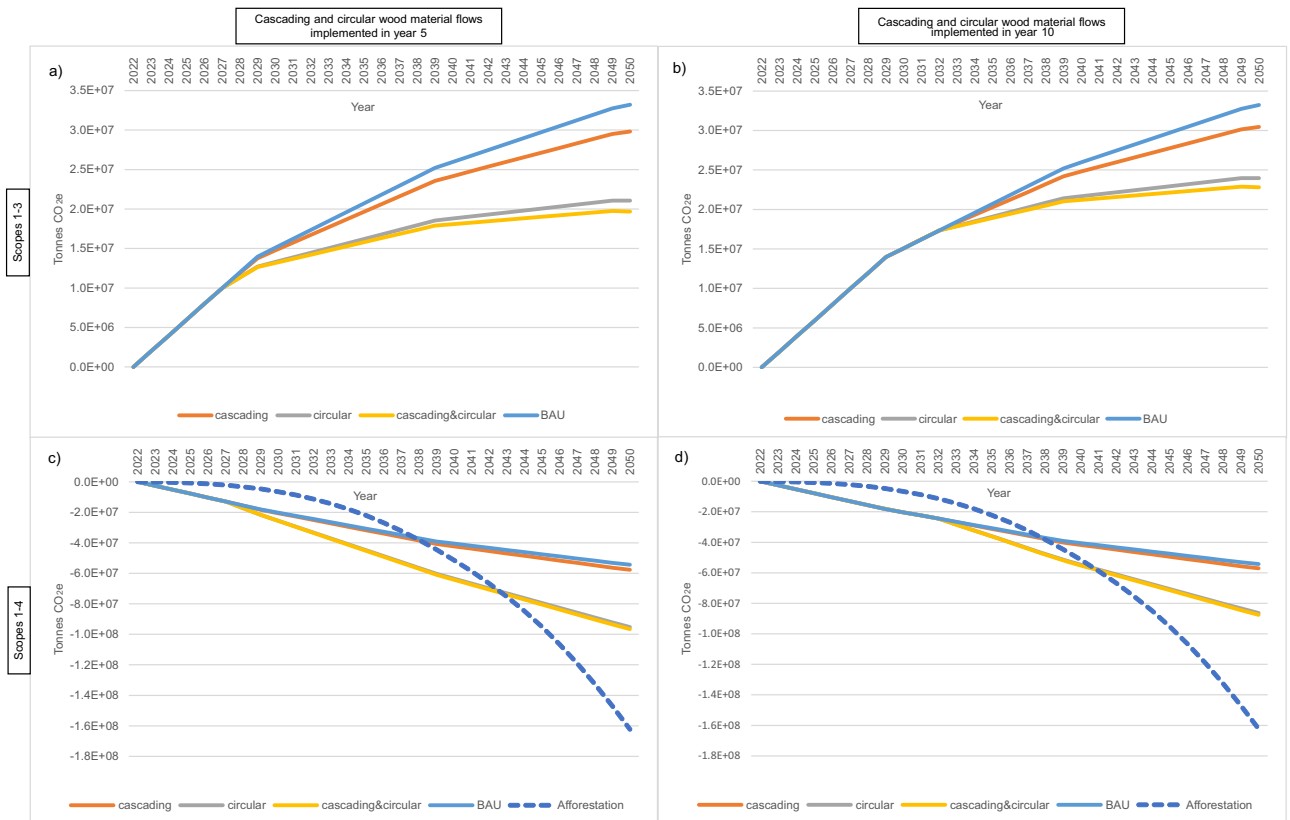

**Fig. 3 | Cumulative global warming potential impact of alternative UK forestry value-chain scenarios.** Global warming potential (GWP) impact is based on a static UK harvest of 9.5 million green tonnes per annum over the study period. Stepwise, decadal decarbonisation is applied in 2030, 2040 and 2050. Graphs allow comparison of net GWP impacts, without (**a**, **b**) and with (**c**, **d**) Scope 4 emissions included to observe the important and changing contribution of Scope 4 (avoided) emissions over time. Graphs also enable comparison of net GWP impacts of implementing circular and cascading scenarios after 5 years (**a**, **c**) and 10 years (**b**, **d**)

to observe the impact of delaying action. Comparing the potential Scope 1–4 GWP impact of the forestry value-chain from existing UK forests alongside the potential effects of a UK national afforestation scenario shows the relative impact and complementarity of these climate-change mitigation strategies. The afforestation scenario assumes a planting rate of 20,000 ha per annum (50% commercial conifer forest and 50% 50:50 conifer:broadleaf forest) from 2023 to 2050. Source data are provided as a Source Data file.

(Table 1). Overall, we observed low awareness of the potential role of forestry value chains in a CE.

Adequate financial and personnel resources are essential for change agents to be able to perform their role, individually and collectively[20]. However, shared 'resourcing' barriers are identified by participants from across the value chain, with low operating profit margins and uncertain future wood supply concerns dominating; the latter is compounded by uncertain government land-use subsidies, a convoluted woodland-creation approval process and unpredictable revenue from voluntary carbon markets impeding afforestation. Participants reported that these barriers delay stakeholder decision-making, restrict the ability or willingness to invest in change initiatives, and impede recruitment and retention of skilled labour (Table 2).

A change system requires a social dynamic that motivates and drives change[20]. However, we found that value-chain stakeholders feel little social pressure to change from either industry peers or consumers—perhaps reflecting an apparent absence of change stewards 'advocating' for circular wood value chains. We also found that conservatism towards change and innovation limits 'prototyping' of new technologies and also new ways of organising, new policies, new financial products, and ideas to influence consumption. Few examples of large-scale CE demonstration projects have materialised to date (Table 2).

In order to appraise the wider GWP impact and circularity of the forestry value-chain, more holistic 'measuring' is needed at the value-chain level to benchmark, reveal opportunities for improvement and

then monitor improvements[20] toward implementation of CE principles. Participants reported that limited transparency of "waste" wood flows is a barrier to effective value-chain measuring that limits the development of waste-wood markets. There is also a lack of practicable metrics for stakeholders at an organisational level to quantify baseline GWP and circularity performance and to monitor progress (Table 2).

## Change initiatives for progress

A shared CE vision for the forestry value chain, functioning as an effective SCS, needs to be agreed upon internationally (Table 1). An effective SCS forestry value-chain could incorporate climate-smart forestry[26,27], cascading wood uses[28,29], "cyclical materials flows, renewable energy sources and cascading energy flows... to limit the throughput flow of materials and energy to a level that nature tolerates..., respecting their natural reproduction rates"[5].

Even with a shared CE vision, coherent-convergent action across the forestry value chain is challenging because of the diverse sub-sectors and scales of businesses, from owner-operators to subsidiaries of global corporations. Trade organisations could play an important organising and influencing role[20] in transitioning the forestry value chain towards an effective SCS. However, despite the many shared barriers to change reported across the diverse stakeholders interviewed, there remains a lack of collaboration across numerous trade bodies representing industry sub-sectors. Trade bodies could become multi-stakeholder 'bridging leaders'[30,31] of change by aligning in their individual[32] efforts towards the shared vision (Table 1 and Fig. 4).

**Table 1 | Societal change system barriers and enablers**

| Function | Barriers to change | Enablers to change |
|---|---|---|
| System visioning | No unifying (global) vision for the role of forestry in a net zero circular economy.<br>Vision has narrow focus on decarbonisation.<br>Weak or no presence of circular economy principles applied to forestry value chains in vision. | Develop a unifying global vision for forestry in a circular economy (CE) by an international coalition of respected forestry and wood organisations (reflecting the global nature of HWP trade) to guide the development of localised (regional, national and industry) vision.<br>Develop a national roadmap, led by trade organisations, defining CE vision and principles to guide action nationally. |
| System organising | Limited willingness to collaborate across industry organisations and networks and other initiatives, leading to gaps in effort and unquantified missed opportunities (e.g. biotechnology and silviculture).<br>Incoherent policy relating to circular material use. Hindered by frequent turnover of politicians.<br>Inadequate waste-wood sorting system.<br>Fragmented land ownership and use.<br>No centralised coordination of multiple small operators in timber haulage. | Greater engagement and collaboration between sub-groups of the forestry value chain. Unite trade associations under a coalition change initiative to work coherently towards an agreed shared vision.<br>Develop and agree on a value-chain CE action plan, including decarbonisation and circularity targets and a strategy for transformation.<br>Map existing change initiatives to identify gaps or duplications in efforts, as well as opportunities for synergies.<br>Establish change steward(s) to create spaces, encounters, and supporting relationships between change initiatives to reveal and address gaps in effort, unproductive duplication and competition, and potential synergies (in each sub-system—with coordination across sub-systems).<br>Enhance organisation of waste-wood recovery and sorting system to enable the development of recovered wood markets: define a coherent national policy that incentivises circularity initiatives (and removes conflicting policy, e.g. biomass incentives); reinforce with supporting waste regulation. |
| Learning | Low awareness and knowledge of CE, specifically the role and potential impact of the forestry value-chain within it, and its implications for decarbonisation. Production-system thinking is prevalent in the forestry value chain rather than change-system thinking.<br>New information/research is slow to be made public (but subsequently disseminated quickly).<br>Gaps in research and/or knowledge sharing—manifest as lack of knowledge for alternative commercial species to Sitka spruce, of their silvicultural characteristics and commercial wood properties (for new bio-products); and the effect of different silvicultural techniques on carbon stocks in (different types of) soil. | Form a collaborative organisation(s) that acts as a change steward(s) to help develop a new 'change-system' mindset through all levels of the value chain.<br>More innovation and less conservatism by businesses across the value chain.<br>More collaborative research and knowledge-sharing initiatives across the value chain.<br>Better quantity and quality of evidence from research, in particular from better "evidence synthesis" across different research studies. |

Matrix analysis of perceived barriers to circularity and decarbonisation in the UK forestry value-chain, identified in stakeholder interview responses. Barriers are categorised according to the system functions 'system visioning', 'system organising' and 'learning', which are needed for effective change[20]. Potential counter-enablers are subsequently proposed to overcome the identified barriers.

We suggest that an international group of progressive forestry and wood organisations collaborate to define a shared long-term strategic vision for the role of the forestry value chain in delivering a net-zero CE and develop a roadmap (in consultation with value chain stakeholders) to guide coherent–convergent action, identifying key opportunities and enablers for change, based on scientific evidence. This is foundational for effective change. The next critical step is consistent, widespread advocacy of the vision and roadmap, in order to create energy for change and turn aspiration into action. Change stewards, including trade bodies, must advocate within the forestry value chain to drive collaboration, knowledge-sharing and innovation (supported by creating spaces for stakeholders to collaborate and exchange ideas); and outside the forestry value chain to lobby the government for coherent supporting policies across relevant domains (agriculture/land use, built environment, waste management, climate and environment, energy). Unity of message, aligned to the shared vision, is critical.

Since our LCA analysis provides clear evidence of the climate-change mitigation benefits of circular and cascading forestry value chains, mandates or incentives to recycle waste wood could represent critical control points to maximise climate-change mitigation arising from commercial forestry. Mandating detailed reporting of wood flows—particularly recovered wood use—could reveal opportunities for CE initiatives (such as MDF recycling, recycled-MDF production and increasing sawn wood production) and enable measuring, target-setting and monitoring of progress. It could facilitate enforcement of higher wood recycling rates, as well as broader implementation of extended producer responsibility[33] to drive recyclability of HWP by making producers responsible for management of their products when they become waste. Mandatory decommissioning plans at the design phase of construction projects over a threshold value; along with a mandatory materials inventory (including technical specifications, such as timber grade) post-construction phase, could drive up recoverability and recycling of used construction materials. However, to ensure the true development of CE at a global level and to prevent leakage, there also needs to be strong governance on the use of biomass for bioenergy. A key benefit of circular wood use is reduced demand for virgin wood (Figs. 1–3), which would be undermined if the use of virgin wood for bioenergy increased to replace recycled waste wood (fuel).

Creating a funding mechanism for implementing and scaling-up circular economy initiatives in forestry value chains is needed to overcome resourcing barriers in this financially constrained sector. Forestry value-chain businesses are not directly credited for most emissions-reduction or carbon-sequestration gains from improved circularity (Fig. 1) and they will not be motivated or have sufficient resources to invest in operational or structural changes without financial support.

Finally, simplifying and accelerating the planning approval process for productive forest planting would reduce costs, complexity and delays to afforestation. These policy changes would also convey public support for productive forestry, indirectly enhancing staff recruitment prospects and the growth of the sector. This will help to ensure the longevity of domestic wood supply and the ability to meet future demand sustainably, as well as providing important carbon sequestration in the short- and long-term.

In conclusion, we present new evidence substantiating the climate-change mitigation benefit of organising more circular forestry value chains and demonstrate how such value chains could interact with commercial afforestation to deliver immediate and sustained decarbonisation. Using UK examples, we show that implementing 'cascading and circular' wood use could deliver 78% more cumulative

**Table 2 | Societal change system barriers and enablers**

| Function | Barriers to change | Enablers to change |
|---|---|---|
| Resourcing | Uncertainty of land-use subsidies and voluntary carbon market prices delays action.<br>Investment risk limits investment. The risk is due to reducing UK harvest volumes, lengthy and expensive woodland creation planning applications, uncertain land-use subsidies and voluntary carbon market prices.<br>Low cost-competitiveness (of wood/bio-products vs alternatives)<br>Low-profit margins, which limit business reinvestment and recruitment.<br>Limited government support for decarbonisation initiatives.<br>Insufficient price differential between higher (carcass) and lower (fencing) value sawn wood products to incentivise hierarchical cascading use, and the insufficient or unfavourable price differential between HWP and non-wood substitutes to favour HWP. | Increase government commitment and support for commercial woodland creation schemes. Provide clarity on subsidies for woodland creation (environmental land management scheme (ELMS) in England and equivalent schemes in the other UK nations).<br>Develop a voluntary carbon standard recognising the contribution of HWP to decarbonisation and the CE.<br>Simplify the planning application system for woodland creation.<br>Increase government financial support for effective CE and decarbonisation change initiatives.<br>Influence relative pricing of wood-based products to reflect holistic value in a CE, i.e. reward resource-use efficiency and low embodied carbon, via differential government subsidies or regulatory barriers. |
| Measuring | Poor transparency of material flows through the value chain. Particularly poor transparency of flow of recovered wood to cascading and end-of-life uses.<br>No widely agreed circularity metrics.<br>Low participation in monitoring of emissions, particularly by SMEs. | Regulation for mandatory reporting of wood flows through the value chain, in particular for waste wood.<br>Development and application of practical circularity metrics at organisation and value-chain levels.<br>Calculating, reporting and monitoring of GHG emissions at organisation and value-chain levels.<br>Development of value-chain circularity and decarbonisation targets in the form of an industry road map and transformation strategy document.<br>Regulation for mandatory materials-inventory for new construction projects (and maintained by the asset owner over the structure's life in order to facilitate recovery and recycling at end-of-life). |
| Advocating | No prevalent change stewards applying pressure on the value-chain to transition towards decarbonisation and circularity across the value-chain.<br>Low social pressure felt from industrial, commercial and public consumers of wood – due to a lack of awareness and apparent interest. | Create a collaborative organisation(s) that acts as a change steward(s) to take on advocacy roles, including lobbying of government to implement supporting policy and regulation.<br>A community of value-chain stakeholders advocating for CE system change within their professional networks (led by principles set out by the change steward(s) and roadmap).<br>Organisations set and declare internal decarbonisation and CE targets and request suppliers to do the same.<br>Communicate change initiatives and successes widely to increase social pressure and build energy for change.<br>Engage and collaborate with existing impactful CE advocators (e.g. Ellen MacArthur Foundation[4]). |
| Prototyping | There is conservatism towards change. Interviewees reported a lack of prototyping to support a shift to net-zero CE, such as new ways of organising, policies, financial products and ways of influencing consumption. Few large-scale demonstration projects. | Stakeholders, acting individually and collaboratively from across the value chain (e.g. commercial businesses, academia, consumers, trade organisations, government) to embrace the principles of CE will help evolve a culture compatible with conceiving and implementing innovative initiatives. For example, organisations could integrate innovation into company policy and culture; fund or collaborate on academic research; and engage with emerging businesses and technologies. |

Matrix analysis of perceived barriers to circularity and decarbonisation in the UK forestry value-chain, identified in stakeholder interview responses. Barriers are categorised according to the system functions 'resourcing', 'measuring', 'advocating' and 'prototyping', which are needed for effective change[20]. Potential counter-enablers are subsequently proposed to overcome the identified barriers.

climate-change mitigation by 2050 than the decarbonisation of linear wood value chains alone. Future expansion of this analysis to include currently speculative circular initiatives is likely to identify further mitigation benefits. Circular wood use can reduce Scope 1–3 process emissions, particularly for recycled-MDF manufacturing, though this is not true across all value chains since extra process steps and energy demand can also lead to higher direct emissions[5,34]. Most mitigation is achieved via product substitution and increased carbon storage in HWP and forests (via reduced harvest demand, thereby sparing global forest resources for in-situ carbon storage or other benefits). Due to the international trade of wood products, these impacts will cross geopolitical boundaries. Going beyond business-level emissions' accounting is therefore imperative to realise the significant contribution that a more circular forestry value chain could make towards a net-zero circular economy. Governments can support change at scale by introducing coherent policy, regulation, green procurement and financial incentives.

Lack of organisation across the forestry value chain is impeding a transition towards circularity. There is an urgent need for a globally shared vision of CE forestry value chains to organise collective efforts and create energy for change, leading to coherent-convergent action by all. Consistent advocating will create social pressure[20] and could stimulate broader collective support for the CE vision, inside and outside the value-chain: in policy, finance, technology and consumer domains. Failing to become an effective SCS and implement circularity initiatives heightens the risk of overshooting Paris Agreement-aligned GHG emissions targets, or worse, increasing biogenic carbon emissions due to forest degradation caused by harvest rates rising with global demand for wood, exacerbated by linear wood use.

## Methods
The study was conducted with approval from the College of Environmental Sciences and Engineering—Research Ethics Committee at Bangor University, under the Approval Number COESE2022EF01A.

### Goal of the LCA
Dynamic consequential life cycle assessment[23] was performed on the UK forestry value chain to calculate the GWP impact of four UK wood flow scenarios (Fig. 5) over the period 2022 to 2050 to quantify the potential climate-change mitigation effect of increasing cascading and circular wood uses. We also compared the GWP impact of the four wood flow scenarios against a UK afforestation scenario to benchmark the impact of increasing cascading and circular wood uses against another core, forest-related 'net-zero' GHG emissions strategy.

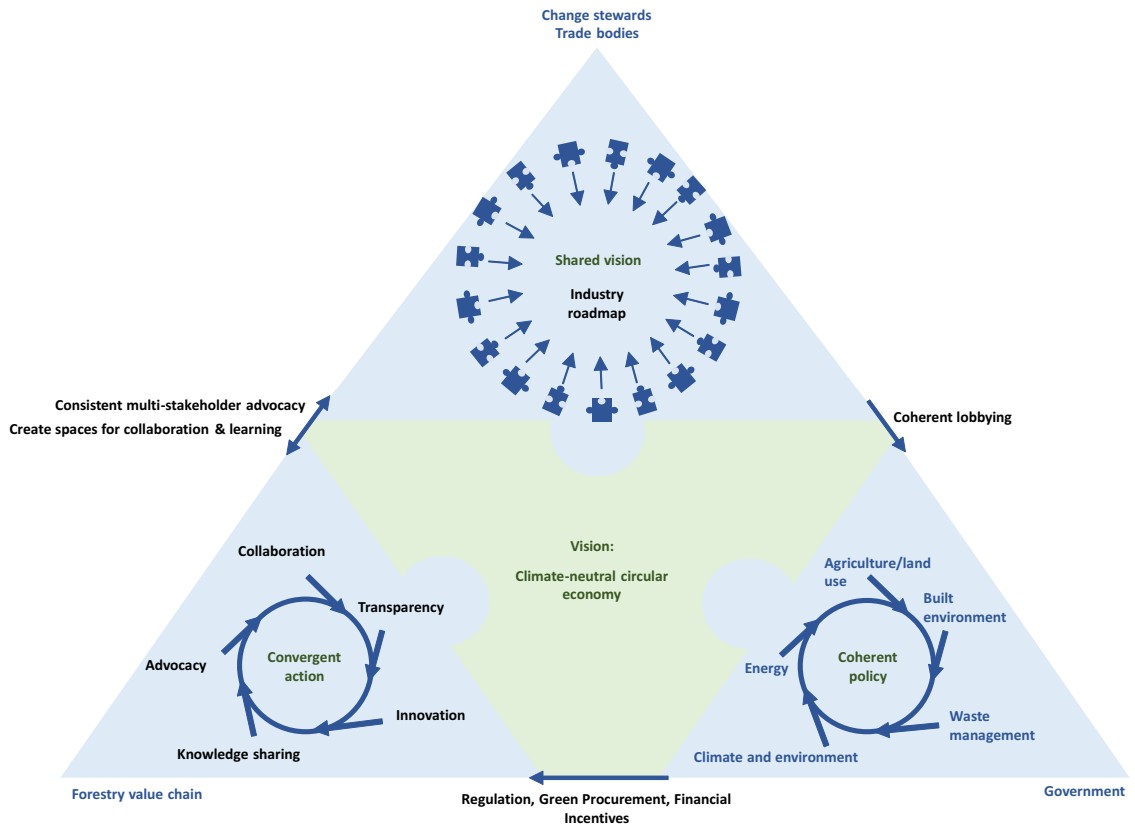

**Fig. 4 | Initiatives to evolve a societal change system.** Interaction is needed between forestry value-chain stakeholders, trade bodies and government to evolve a societal change system that delivers a transition to circular economy principles and optimised climate-change mitigation from the forestry value chain. Black text indicates a call to action.

## Wood flow scenarios

The four wood flow scenarios assessed include business as usual ('BAU') UK wood use (Fig. 5a) and three scenarios representing increased cascading and circular uses of wood (Fig. 5b–d). Full descriptions of these are provided in Fig. 5, along with graphical representations in the form of Sankey diagrams.

The pathways and scenarios we selected for analysis were governed by the stringent data requirements of the rigorous LCA methodology that we have utilised. We carefully surveyed a range of other existing pathways and scenarios, however due to their complexity and major gaps in available data, none of them met the criteria of being suitable for rigorous LCA. This limitation was particularly acute for upcoming technologies. We provide a further assessment of our choice of wood flow scenarios in Supplementary Methods 1 and consider our approach to be conservative with respect to the conclusions derived.

In the circular and cascading scenarios, we minimised direct changes to 'BAU' UK production of (virgin, i.e. not including waste wood) 'woodfuel', 'fence poles' and 'other' HWP as much as possible in order to clearly observe the impacts of the intended key material flow changes, described below. However, due to the complex and dynamic nature of wood flow some changes across UK HWP production volumes were unavoidable. In cascading scenarios (Fig. 5b, d), more material is directed to sawmills, which increases the supply of sawmill residues to woodfuel. We reduced the flow of logs from the forest gate to woodfuel to counter this and to minimise change to the net 'BAU' (virgin) UK woodfuel production.

While the total volume (i.e., domestically produced plus imported) of each HWP type is kept the same in all the scenarios, as the domestic value-chain changes from 'BAU', the import volumes change in each scenario to maintain the total balance. Therefore, in circular and cascading scenarios (Fig. 5b–d), imported HWP volumes (not shown in Fig. 5) adjust in response to changes in UK HWP production in order to maintain constant UK supply, with the exception of increasing imported woodfuel to replace recycled waste MDF in the circular scenarios (Fig. 5c, d). The latter results in a real net reduction in UK woodfuel consumption in circular scenarios.

## Afforestation scenario

The afforestation scenario involves planting 20,000 ha per year from 2022 to 2050, with 50% commercial conifer forest (Sitka spruce) and 50% 50:50 conifer:broadleaf forest (Sitka spruce; Douglas fir; Corsican pine: silver birch; rowan; oak). Harvesting of commercial conifer forest is assumed to commence 50 years after planting, which is beyond the time period considered in this study.

## Scope of the LCA

The LCA scope includes 'UK forestry operations', 'UK wood processing', 'UK HWP C sequestration' and 'UK-bioenergy production' associated with softwood produced from UK forests, except pulpwood (accounting for 5% of UK harvest) (Fig. 6). Pulp and paper manufacturing is highly partitioned from the rest of the value chain and is excluded from the study. Annual UK softwood harvest is assumed to be constant at 9.5 million green tonnes per year[24].

Since we are considering the consequential impact of shifting from 'BAU' wood use and because the annual UK harvest rate is constant in all scenarios, UK 'forest ecosystem' C sequestration is set to

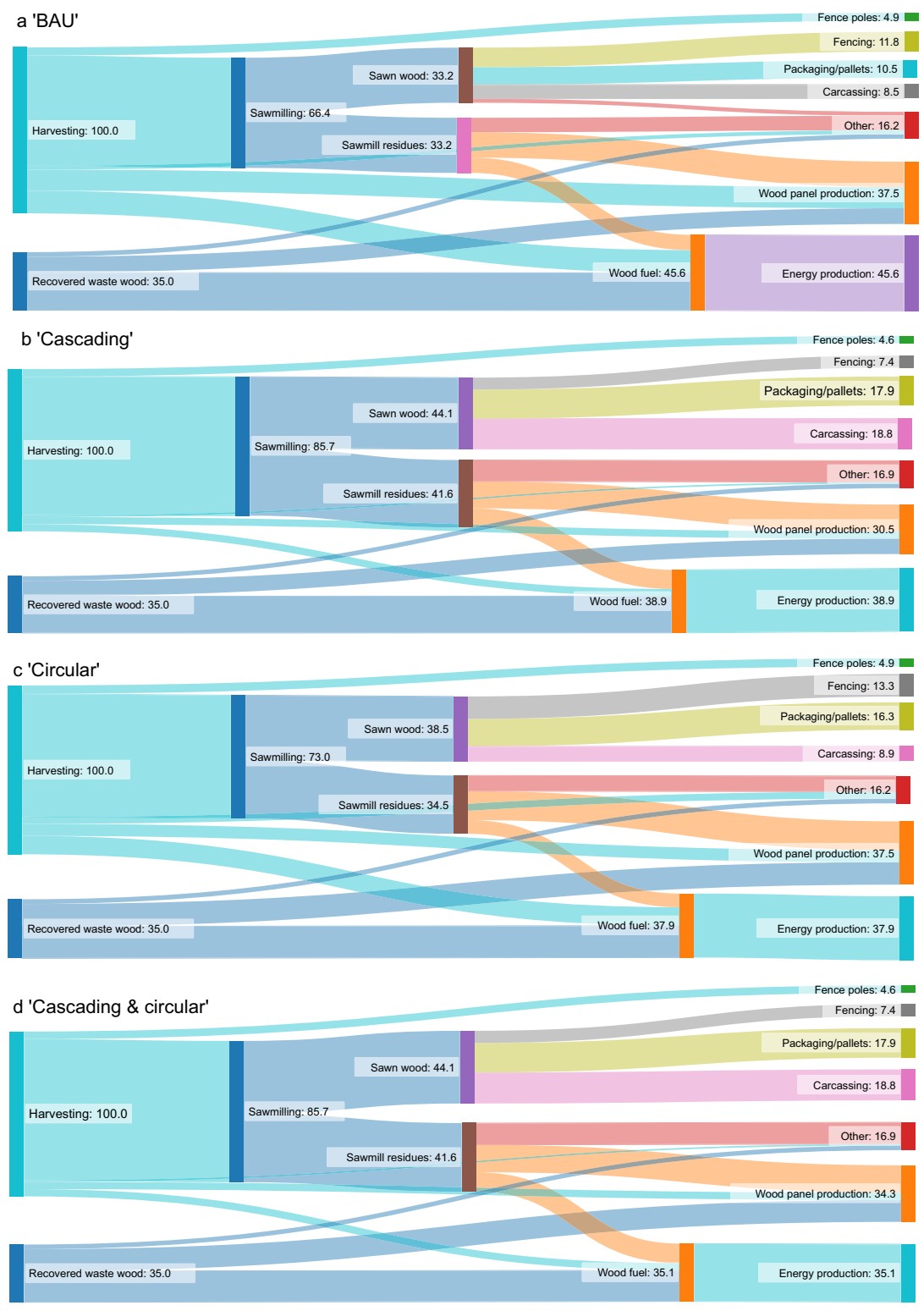

zero in all scenarios. However, for circular scenarios (Fig. 5c, d) that use less virgin wood, gains in forest C sequestration from reducing non-domestic harvesting are calculated ('Avoided emissions−reduced harvest').

HWP C-sequestration benefits are also set to zero for 'BAU', since we are considering the consequential impact of shifting from 'BAU' wood use. 'HWPs C sequestration' accounts for an increase/decrease in the UK HWP C pool relative to the 'BAU' due to changes in 'UK wood

processing' (Fig. 6) in the circular and cascading wood flow scenarios (Fig. 5b–d).

To clearly observe the consequential impacts of increasing cascading and circular wood uses, embodied emissions from processing and transport of 'BAU' imported HWPs are set to zero. The GWP impact of changes to 'BAU' HWP imports (in scenarios 5b–d) is calculated −'Imported HWP processing (change from 'BAU')' and 'Imported HWP transport (change from 'BAU')'.

**Fig. 5 | Wood-flow Sankey diagrams for four modelled UK wood-flow scenarios.**
Flows are scaled to a UK harvest of 100% in order to clearly observe differences
between scenarios. Imported harvested wood products (HWP) are not shown in this
figure but are accounted for in the lifecycle assessment (LCA), as defined in the
methodology text. **a** Business as usual ('BAU') wood flow is UK domestic timber
production, wood processing and waste wood recovery, taken from nationally
reported data[24,50,51]. Note that very little recovered waste wood is landfilled in the
UK, so 'BAU' involves cascading use of recovered waste wood in particleboard
manufacturing and energy production. **b** 'cascading' wood flow arises from a
strategy to increase the conversion of UK harvested wood to carcassing (con-
struction sawnwood). This results in increasing carcass production and decreasing

wood panel production, compared to 'BAU'. By virtue of this improved hierarchical
use, inherent increases in further cascading use occur (carcassing (sawn wood) to
particleboard to bioenergy). **c** 'circular' wood flow arises from a strategy to increase
the recycling of recovered waste wood and involves recycling waste medium-
density fibreboard (MDF) and producing recycled MDF. This results in less (virgin)
harvested wood being used in wood panel production and diversion of 'spared'
material to sawmills, leading to increased conversion of the harvest to sawn wood,
mainly packaging/pallets and fencing. Diverting waste MDF from woodfuel to
recycled MDF also results in lower wood fuel use and energy production.
**d** 'Cascading & circular' is 'cascading' wood flow (**b**) combined with MDF recycling.

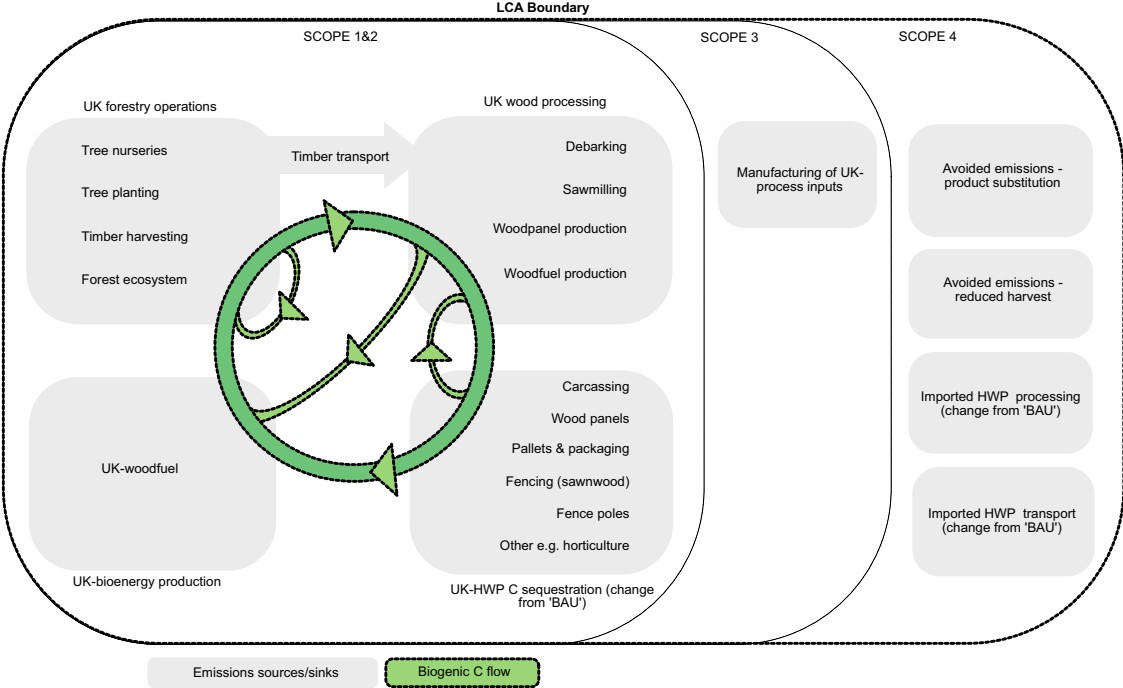

**Fig. 6 | Scope and boundary of the life cycle assessment.** Scope 1 is direct
greenhouse gas (GHG) emissions; Scope 2 is GHG emissions associated with the
generation of electricity, heating/cooling, or steam purchased for own consump-
tion; Scope 3 is indirect GHG emissions other than those covered in Scope 2[48].
Avoided emissions (Scope 4) are GHG emission reductions that occur outside of a

product's life cycle or value chain but as a result of the use of that product[49]. Here,
'BAU' refers to GHG emissions/sequestration from the business-as-usual UK wood
use scenario modelled in the study (represented in Fig. 5). HWP means harvested
wood products.

Results are presented under Scopes 1 & 2, Scope 3 and Scope 4
emissions' categories (defined in Fig. 6) in order to provide further
insight into where emissions arise in the value chain.

### Process emissions
Process GHG emissions are calculated for: 'UK forestry operations'
(including tree seedling production, site preparation, planting, har-
vesting and timber transport (from forest to wood processor); 'UK
sawmilling'; 'UK wood panel production' (particleboard, MDF and
recycled-MDF)); 'UK woodfuel production'; 'imported-HWP proces-
sing' (change from 'BAU') and 'imported-HWP transport' (change from
'BAU'). GHG emissions are calculated from secondary data (Ecoinvent
v.3.539 using OpenLCA v1.7.4), scaled to the four material flows (Fig. 5).
The production and transport of all material and energy inputs were
accounted for, as were the construction or manufacture of infra-
structure and capital equipment. Full life cycle inventories (and impact
calculations) are provided in Supplementary Data 1–4 and Source Data,
with an example inventory table for the 'BAU' wood-flow scenario in
Supplementary Table 1. Given that the focus of this paper is on climate-
change mitigation, only the global warming potential (IPCC 2013 GWP

100a[35]) impact category was evaluated, expressed as kg $CO_2$e. The LCA
scope includes direct and indirect GHG emissions from wood pro-
duction, processing, transport and use (Fig. 6). The GHG emissions
from wood processing are treated as the same per unit product for
equivalent UK- and imported-HWPs, on the basis that most sawmills
use wood residues for heat energy and assuming similar average
electricity generation emissions and sawmill efficiency across domi-
nant source countries for UK softwood imports.

### Terrestrial carbon
Harvest reduction due to recycling MDF (leading to 'Avoided emis-
sions—reduced harvest') was calculated as the volume of recycled
MDF converted to green tonnes equivalent using conversion factors
developed by Forest Research[36]. We assume that harvest reduction
occurs in countries from which imported-HWP are exported, but
model forest growth of commercial forest that is typical of UK sys-
tems as a proxy for this international effect. Specifically, we assume
that a Sitka spruce forest equal to the total area of UK commercial
forest, 710,000 ha, shifts from a 50-year harvest rotation, to a 54-year
rotation – with the resulting annual harvest reducing by the defined

amount. 'Avoided emissions−reduced harvest' is calculated as the annualised, 28-year (study period) average gain in forest ecosystem (soil and biomass) carbon as a result of shifting to the longer rotation.

Forest growth, decay and harvest volumes were calculated using the Carbon Budget Model for the Canadian Forest Sector (CBM-CFS3)[37], which complies with carbon estimation methods outlined in the 2003 Intergovernmental Panel on Climate Change (IPCC) Good Practice Guidance For Land Use, Land-Use Change and Forestry[38], and the 2006 IPCC Guidelines for National GHG Inventories[39]. It was parameterised using best-fit yield tables from Forest Yield, the standard yield model for forest management in the UK[40]. For mixed-species forests (defined in the afforestation scenario), aggregate group yield classes (YC) were calculated based on weighted mean YCs. CBM-CFS3 outputs include annual soil and biomass carbon stocks. Calculations thereby account for the decline in annual increment as trees age past the standard rotation age in 'circular' scenarios of 'Avoided emissions−reduced harvest'.

### HWP carbon storage

Retirement rates of HWP were calculated according to the IPCC[41] simple decay approach. Thus, there is a carbon transfer from domestic forest carbon pools to HWP at the point of harvest, which is emitted from the HWP pool at the time of end-of-life of that HWP. We account for the annual release of carbon to the atmosphere from HWP (including fuelwood), where wood came from domestic harvest. We do not account for imported HWP carbon storage because this is credited back to the exporting countries and would not change in the absence of UK imports. We use IPCC[42] and modified HWP decay factors[43] to calculate HWP retirement emissions. Emissions from landfill disposal of retired HWP (very low in the UK[6]) are excluded. 'UK-HWP C sequestration (change from BAU)' is calculated as the annualised, 50-year average gain in UK-HWP C as a result of change to UK HWP production in cascading and circular scenarios (Fig. 5b–d) compared to 'BAU'.

### Substitution credits

Substitution credits were calculated following the same method as Forster et al.[6] and summarised in the following text. Fuel-to-energy conversion factors (for natural gas and wood chips) were taken from Ecoinvent data (Ecoinvent v.3.539 using OpenLCA v1.7.4) unit processes[44] to calculate fossil fuel (natural gas) substitution by dedicated biomass energy generation and incineration with energy recovery for wood waste. Emissions avoidance through the substitution of mineral construction materials was estimated by translating the final mass of construction timber (150 tonnes at 20% moisture per hectare of the thinned forest) into an area of the timber-framed wall using industry-standard design: 0.0175 $m^3$ of timber per 1 $m^2$ wall (BRE IMPACT database[45] accessed via eToolLCD® software). 1 $m^2$ of timber frame wall replaces 1 $m^2$ of single skin, 140-mm concrete block and mortar (sand:cement ratio 10:3) wall with 10-mm jointing in typical UK house construction. Avoided emissions were then calculated using emission factors from Ecoinvent for the manufacture of concrete blocks, sand and cement. Substitution credits are subject to progressive decarbonisation factors, outlined in the following section.

### Progressive industrial decarbonisation

Projected industrial decarbonisation factors are applied to all process emissions across forestry operations, wood processing and transport—for domestic and imported HWP. They are also applied to substitution credits for avoided (natural gas) electricity generation and concrete production. Decarbonisation assumptions are elaborated in Supplementary Methods 2. Bioenergy with carbon capture and storage (BECCS) deployment is not accounted for, owing to the short time horizon of the study and unpredictable timeline for BECCS technology readiness.

### SCS analysis

We captured experiential knowledge of key stakeholders operating in the forestry value chain to identify perceived barriers to CE and then performed SCS analysis to understand the value chain's capacity for change.

### Participants

Fifteen individuals from the membership of the UK Confederation of Forest Industries (Confor) were selected, in consultation with Confor senior leadership, to participate in semi-structured, one-to-one interviews; thirteen agreed to take part. Four participants from outside Confor's membership were also invited in order to strengthen representation of sub-sectors not prevalent in the membership base, including wood recycling and wood panel manufacturing. We were unable to recruit a participant from the forestry investment sector. Seventeen individuals were interviewed in total. Informed consent was obtained from all participants.

Participants were selected for their level of experience (seniority), area of expertise (sub-sector) and spread of geographical location (to capture variation in experiences from across the UK). Participants mostly represented private sector organisations, but public forestry organisations were also included. Interviews took place online via Microsoft Teams or by telephone. Most interviews lasted between 45 and 60 min, with five lasting 30–45 min. Fourteen were video and audio recorded (with automated transcription). Three were not recorded and notes were taken manually.

### Interviews

Participants were asked about decarbonisation and circularity initiatives in their business to determine the experiences (successes and challenges) of each interviewee's organisation with regard to energy use, energy reduction, carbon reduction and wood-use efficiency. They were then asked broader questions about their sub-sector and the whole value chain regarding barriers and enablers of change towards decarbonisation and circular economy.

### Analysis of interview content

Directed content analysis[46,47] was applied to whole-interview video recordings and manual notes using a combination of deductive and inductive approaches[47]. First, we developed an analysis matrix based on the five change sub-systems (column headers) and seven change functions of an effective change system (row headers) to organise the interview data. Synthesised and anonymised interview data is provided in S9.

The five change sub-systems are Technology−research organisations and companies developing new technologies and innovations; Policy−governmental bodies, including regulators and legislators, and other stakeholders that engage in co-production or influencing of rules and policies; Producer−the infrastructure that produces, processes and distributes wood-based products; Consumer−demand for wood-based products and the influence of demand; and Finance−public and private sector capital and organisations innovating in and influencing financial markets.

Every change sub-system must perform the following seven change functions for an effective change system. System Visioning−a shared vision that creates coherence among stakeholders and changes initiatives; System Organising−organising of effort and stakeholders in ways that provide coherent aggregation of voices at scale; Resourcing −provision of financial and personnel resources needed for action; Learning−development and exchange of knowledge arising from prototyping; Prototyping−Developing and testing of new technologies, ways of organising, policies, financial products and ideas; Measuring−assessing progress towards the vision, and identifying opportunities for improvement; and Advocating−social pressure and energy for change. (Definitions adapted from Waddell[20]).

We synthesised the interview data for the five sub-systems into one whole system, with the barriers still categorised under the seven change functions. We then followed inductive analysis principles to develop further categories within the bounds of the seven change functions in the matrix to describe and group the barriers to change. For example, within the 'system organising vision' function category, the following sub-categories were created to group and describe the barriers reported by participants within this function: 'limited willingness to collaborate', 'incoherent policy', 'poor waste sorting system', 'fragmented land ownership' and 'no-centralised co-ordination of multiple small operators.' This was performed for all seven change-function categories. The concepts discussed were complex and sometimes interconnected, so manual coding was applied to avoid the risk of missing relevant information.

## Data availability

Source data are provided in this paper. The data generated in this study are provided in the Supplementary Information and Source Data files. Background data were generated using the publicly available CBM-CFS3 model and extracted from the Ecoinvent v.3.539 database. All subsequent calculations were undertaken using standard MS Excel functions. Source data are provided in this paper.

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

## Acknowledgements

This work was funded by the UK Natural Environment Research Council (NERC) Envision Doctoral Training Program NE/L002604/1 (awarded to E.F.). Additional CASE studentship funding was provided (to E.F.) by Woodknowledge Wales (Ffarm Moelyci, Felin Hen Road, Tregarth, LL57 4BB, UK) and Coed Cymru (The Forest Hub, Unit 6, Dyfi Eco Park, Machynlleth, Powys, SY20 8AX, UK). The authors are grateful to all interviewees for taking part in this study and to Stuart Goodall from the Confederation of Forest Industries (Confor) for his support in identifying and recruiting interviewees.

## Author contributions

E.F. led the original conception of the study, undertook data collection and analysis, and drafted the paper. D.S. and J.H. participated in the original conception of the study, informed the study design and edited the manuscript. G.N. participated in the original conception of the study and edited the paper.

## Competing interests

The authors declare no competing interests.
