## [Peer Review File · Nature Communications]

REVIEWER COMMENTS

Reviewer #1 (Remarks to the Author):

The current work by Forster et al., for incorporation of CE approaches via cascading and circular use of wood to mitigate the GWP impacts along with a discussion on implementing a societal change system suggest interesting results in context to forestry value chain in UK. The authors performed LCA and conducted semi-structured interviews to screen the barriers present in the implementation of circular economy strategies suggested, which is an intriguing approach and could be fruitful reference for the stakeholders. However, some concerns with the current manuscript are:

1. The authors have kept the study focused to UK. However, how do they think that the conclusions derived from this work will apply for the global context?
2. In the paper, the Figure numbering is not sequentially placed. Please modify it.
3. Authors have taken into consideration of MDF. What about the engineering wood and other upcoming technologies in the market? What are their potential effect by the year 2035?
4. Figure 2 and 3 needs significant editing in terms of utilization of appropriate legends and subscripts. The 2 in Co2 should be in subscript.
5. In the study authors have used the word wood as a generalized statement and have not categorized it properly. Does the variety types of wood for e.g hardwood soft wood etc have any effect on the study? What are the required assumptions one needs to consider?
6. Authors need to rewrite the abstract and include the relative benefits of CE than current BAU system (%).
7. Please discuss the disadvantages of the cross-sectoral integration of forest value chains in terms of technical, social, and economic reforms in forestry value chain.
8. The finance associated with implementing the CE approaches in forest value chain would be an important practical factor. Many industries prioritize the economical benefit over sustainability of the sector. It is thus suggested that a technoeconomic analysis or cost analysis of the whole systems for implementing CE approaches can be conducted and compared with BAU with decarbonization, to quantify the exact value required for the reforms.
9. Authors in this work only suggest the policy-based changes needed to implement to achieve CE and discussion on the technical implementation with examples are lacking. Authors are therefore suggested to present brief discussion of more CE scenarios connecting different industries for waste reduction.
10. One of the study parameters include the consideration of UK, as wood flow remains within the national boundaries and traceability is high. Conducting the same study in a developing nation with more exports would be a complex system to study. Please discuss more on the rational of taking only UK as the nation for current study. Please also comment on the implications of the current approach in case of a developing nation.

11. UK domestic forestry value-chain relies heavily on imports. The HWP imported from other countries might be adding to more GWP emission that is being mitigated in UK by employing CE approaches. Moreover, in my opinion, the impacts of HWP imported, has not been discussed effectively. How about if other countries also start avoiding emission by relying on HWP from different nations.

12. The redirection of harvested wood in sawn wood and wood recycled medium density fiberboard (rMDF), do these products put any effect on the technical processing and socioeconomical changes in the respective sectors?

13. Authors also need to briefly discuss climate change parameters other than GWP. Also, is the data presented in normalized form? Authors are also suggested to study the variation in impact values if source of electricity is varied (from hard coal to natural gas).

14. The process of avoiding the emission by substituting the products is advocated to contribute the greatest GWP credit. The substitution of fossil fuel by wood fuel will still contribute to GWP impacts and might not be the best sustainable strategy for substitution of fuel sources. What would be the effect of substituting fossil fuels with renewable eco-friendly energy sources in this study? Similarly, substitution of concrete by sawn wood is also a less robust system comparatively. Utilization of forest biomass as commercial construction materials still has a long way to go and may require continuous maintenance.

15. Authors need to comment on the rationality for these avoided emissions by product substitution examples. At the same time, it is encouraged to propose more product substitution strategies.

16. Implementation of cascading and circular wood materials flows in year 5 shows a higher rate of GWP impact decrease than implementation at year 10. However, both the implementation reaches the similar GWP value by the end of the time frame used for study period, i.e., 28 years. Which period of implementation would be better then? Please comment.

17. Decarbonization in BAU approach is implemented in stepped decadal decarbonization strategy. When compared with the cascading or circular approaches, the BAU with decarbonization only remains 5×10^5 CO₂ i.e., 25% of initial CO₂. I think instead of using cascading, circular, and both in combination, if only BAU with decarbonization is applied, then the current reform for diverting the wood flows, technical processing, and implementation of strategies would not be required. If the study period has been taken for more than 28 years, the BAU as well has reached to 0 CO₂ emission by the end of study period.

18. Pulp and paper industry is an important component of the forest value chains. Please comment on the rationality of its exclusion from the study.

19. The resolution of all the figures needs to be improved and formatted (CO₂ and not CO₂) as per the journal guidelines.

20. The authors need to check the grammatical errors, language, and formatting errors in the manuscript.

21. Line 393: "Potential climate mitigation effect" what does it mean?

22. Authors need to format the referencing properly. The current form of referencing is very poor and not as per journal guidelines.

Reviewer #2 (Remarks to the Author):

The manuscript entitled “Circular wood use can accelerate global decarbonisation but requires cross-sectoral coordination” identified the potential of carbon storage and greenhouse gas reduction in the UK wood-based value chain. This study covers cascading and circular wood use and afforestation as strategies to contribute in the Paris Agreement targets achievement based on a dynamic lifecycle assessment (LCA). For such, the authors presented barriers based on stakeholder interviews and proposed enablers to plan a and societal change system (SCS). To the best of my knowledge, this is a novel research that combines LCA and SCS to support a robust decision making in the wood sector, which can be applied in different contexts. In addition, the data are presented in the Supporting Information for transparency. Therefore, I recommend the publication of this manuscript.

Reviewer #3 (Remarks to the Author):

Dear authors

The manuscript provides a significant scientific contribution to devising future climate mitigation strategies for forestry-based sectors. However, here are a few comments that need to be addressed.

1. The term 'cascading scenario' is misleading. Cascading is the sequential use of wood in multiple applications. In your case, you have considered only 1 aspect of cascading: prioritising primary wood use for high-value applications (i.e. sawn wood). You refer to secondary wood use as 'circular'. I think the terminologies could be adapted to avoid confusion.

A relevant reference in this regard is - Navare, K., Muys, B., Vrancken, K. C. & Van Acker, K. Circular economy monitoring – How to make it apt for biological cycles? *Resour. Conserv. Recycl.* 170, 105563 (2021).

2. You emphasise the need for cross-sectoral integration of sustainability assessment. However, your scenarios are centred around wood use. Forestry is included only by considering 'avoided harvest' and increased afforestation. Both refer to the amount harvested but not to other forest management strategies (e.g. increasing forest diversity, prioritising plantation of certain types of wood etc.)

3. You have assumed that reducing the harvest increases the carbon stock in the forest. Is that always the case? It depends on the state of the forests - leaving the additional biomass in the forest may decrease the overall sequestration over time as emphasised in Nabuurs, G. J. et al. First signs of carbon sink saturation in European forest biomass. *Nat. Clim. Chang.* 3, 792–796 (2013). It is essential to account for this in your analysis (Of course, this is keeping aside the other ecosystem services).

4. [Line 410 - 414] Could you specify the wood imports for different scenarios? I understand that import amounts remain the same, however, the product that is imported is different in each scenario. Could you make it explicit in the figure or the text?

RESPONSE TO REVIEWER COMMENTS

We sincerely thank the Reviewers for the valuable time and thought they have committed to reviewing our paper. We have carefully considered every individual comment and set out our responses to these point-by-point below, including descriptions of any resulting modifications to our manuscript and supporting information.

Reviewer #1 (Remarks to the Author):

The current work by Forster et al., for incorporation of CE approaches via cascading and circular use of wood to mitigate the GWP impacts along with a discussion on implementing a societal change system suggest interesting results in context to forestry value chain in UK. The authors performed LCA and conducted semi-structured interviews to screen the barriers present in the implementation of circular economy strategies suggested, which is an intriguing approach and could be fruitful reference for the stakeholders. However, some concerns with the current manuscript are:

1. The authors have kept the study focused to UK. However, how do they think that the conclusions derived from this work will apply for the global context?

We previously alluded to this in our Conclusion and have made further edits to the “Analysis of wood use strategies in a UK context” and Conclusion sections to make this more explicit. In essence, this study provides a deep dive into some example value chain configurations and barriers and enablers to more circular use of wood, requiring detailed information that can only realistically be collected at a national level.

2. In the paper, the Figure numbering is not sequentially placed. Please modify it.

We have checked the Figure numbering and believe it to be in the correct sequence with the first citation of each figure in the text.

3. Authors have taken into consideration of MDF. What about the engineering wood and other upcoming technologies in the market? What are their potential effect by the year 2035?

We acknowledge that there are many more prospective cascading and circular pathways that would benefit from further study. Indeed, this is an important finding of our paper, which we highlight. We emphasised the need to better understand the limitations and technical feasibility of new upcoming technologies, including the important data gaps we identified regarding transparency of recovered wood product flows (both volume and quality). Our selection of pathways and scenarios for analysis was governed by the stringent data requirements of the rigorous LCA methodology that we have utilised. We carefully surveyed the range of other existing pathways and scenarios, however due to their complexity and major gaps in available data, none of them met the criteria of being suitable for rigorous LCA. This limitation was particularly acute for upcoming technologies. We have added a summary of this survey of other pathways and products to the Supplementary Information (S10).

To match the research objectives of the paper, we selected recycled MDF as the circular case study because it is an almost closed-loop process (hence it is a circular, rather than cascading, use), and because sufficient data were available to meet the requirements for rigorous LCA. Specifically:

- 1) there is data available on the annual volume of recovered MDF at the UK national scale of our study
- 2) the technology is proven and is being commercially scaled-up, and we could therefore obtain lifecycle inventory data for the process.

Furthermore, MDF recycling is a conservative representation of circular use in general because it does not involve product substitution credits, just harvested wood product (HWP) C-storage credits. It also has a moderate product half-life (20 years) so impacts can be observed in the feasible 100-year timespan of our study. In contrast, upcoming recycling technology pathways (such as bioplastics) will have the potential for material substitution credits, but there is not yet adequate data of these to include in a rigorous LCA.

A different engineered wood product scenario, as suggested by the reviewer, could be that of used sawnwood cascaded into cross-laminated timber (CLT). NB this is a cascading wood use, rather than circular. This could certainly be promising for climate mitigation in the future. However, such a value chain remains uncertain and challenging to implement (and model) as technical barriers remain to be overcome, such as definition of the minimum level of quality and size of recovered sawnwood for this cascading use, which would have major implications for the LCA outcome. Furthermore, there is currently no adequate system in the UK for collecting and reporting data on the quantity (or quality) of recovered wood to provide the necessary data for quantifying the potential implementation at the national scale of our study, and its associated life cycle GWP impact (including accounting for the consequential impacts of any changes to how that wood is currently being used, e.g. diversion from particleboard manufacturing). Inclusion of such a scenario in this study would be highly speculative and would seriously undermine its rigour – but it is certainly something that should be prioritised in future research. We have added an additional supplementary information file, S10, providing our assessment of other potential wood use scenarios (also mentioned in our response to reviewer 1, point 15).

4. Figure 2 and 3 needs significant editing in terms of utilization of appropriate legends and subscripts. The 2 in Co₂ should be in subscript.

We have carefully reviewed these figures making the 2 in CO₂ subscript, and have increased the font size in the legends and other text labels to further increase clarity (in Figures 2, 3 and 5).

5. In the study authors have used the word wood as a generalized statement and have not categorized it properly. Does the variety types of wood for e.g hardwood soft wood etc have any effect on the study? What are the required assumptions one needs to consider?

We have edited the manuscript to clarify that the term 'wood' is used in this paper to refer to softwood (from conifer trees), which is the overwhelmingly dominant wood type used in wood products in the UK.

6. Authors need to rewrite the abstract and include the relative benefits of CE than current BAU system (%).

We have modified the abstract to include key statistics for two important findings from the study. However, we cannot go further than that because we model many scenario variations all of which contribute to the outcome of the study, therefore there is no simple set of 'headline' results that are representative of the study as a whole and that can be presented statistically without lengthy

explanations and caveats. We therefore retain the original abstract text in order to maintain our balanced summary of the key findings of the study.

7. Please discuss the disadvantages of the cross-sectoral integration of forest value chains in terms of technical, social, and economic reforms in forestry value chain.

We make reference to cross-sectoral integration of sustainability policies in our introduction, but cross-sectoral integration of forestry value chains is not in itself a primary recommendation from our study. We advocate for co-ordination of sustainability policy and action, informed by discussions during interviews with value chain actors, and we list identified technical, social and economic challenges of this in Table 1.

8. The finance associated with implementing the CE approaches in forest value chain would be an important practical factor. Many industries prioritize the economical benefit over sustainability of the sector. It is thus suggested that a technoeconomic analysis or cost analysis of the whole systems for implementing CE approaches can be conducted and compared with BAU with decarbonization, to quantify the exact value required for the reforms.

We agree with the reviewer that a technoeconomic or cost analysis would provide an important new insight. However, we are certain that a full cost-benefit analysis carried out with sufficient rigour to be valid, would need to be a whole separate study at least equal in length to the present one. For example, given the very variable timescales for the economic returns (benefits) from the different kind of investments (costs) covered by the options modelled in the present study, it would be essential for a rigorous economic cost-benefit analysis to include scenarios of different discount rates, which would lead to huge complexity in the results. Indeed, this is a very topical consideration given the huge volatility of interest rates in UK and other countries at present. Alternatively, adding a simplistic economic analysis to the present study would seriously undermine its rigour. Furthermore, stakeholders were willing to provide useful detail on barriers and opportunities to change which reflected economic considerations; it is unlikely they would have been so forthcoming with commercially sensitive “hard” economic data. Finally, economic performance of individual actors reflects the market and regulatory environment in which they operate. One of our main recommendations is that this operating environment needs to be changed through coordinated action in order to enhance circularity within wood value chains.

9. Authors in this work only suggest the policy-based changes needed to implement to achieve CE and discussion on the technical implementation with examples are lacking. Authors are therefore suggested to present brief discussion of more CE scenarios connecting different industries for waste reduction.

Within the constraint of the tight journal word limits we are sure that the quality of the paper is maximised by sticking with our well-defined examples for which good data exist in order to demonstrate the potential benefits of enhanced circularity and cascading uses more rigorously. Ultimately there is a large range of prospective alternative scenarios within the scope of industries and waste streams that could be considered, but we can see no valid basis on which to cherry pick additional “examples” from this range (for which we do not have reliable data). Full discussion of the range of possible options would be boundless. However, recognising the importance of alternative scenarios and industries, we have added our assessment of this issue in the SI (S10) including an

overview of the most promising technologies across the range of those currently under development or identified as having good future potential.

10. One of the study parameters include the consideration of UK, as wood flow remains within the national boundaries and traceability is high. Conducting the same study in a developing nation with more exports would be a complex system to study. Please discuss more on the rationale of taking only UK as the nation for current study. Please also comment on the implications of the current approach in case of a developing nation.

In order to be valid, dynamic LCA requires material flow data of sufficient reliability, which was a major factor in our decision about what national frame to use for our study. The same approach can be applied to any nation for which sufficient data are available – in particular data of material flows (of wood through the value chain). This would present a serious challenge or limitation for application to a developing nation if such data gaps exist. In such cases, very different analytical methods would need to be used to provide more approximate estimations of net global warming impacts, but they lie outside the frame of the present study. In this context an important outcome of our study of high relevance to developing nations is to emphasise the need for good traceability of wood flows to enable more circular and cascading uses of wood.

11. UK domestic forestry value-chain relies heavily on imports. The HWP imported from other countries might be adding to more GWP emission that is being mitigated in UK by employing CE approaches. Moreover, in my opinion, the impacts of HWP imported, has not been discussed effectively. How about if other countries also start avoiding emission by relying on HWP from different nations.

The reviewer makes an important point here. However, we are confident that we have addressed this point as fully as can be justified based on the evidence provided by our results.

We agree that the UK's import dependency could have climate change mitigation consequences in other countries, especially through increased harvest pressures, and conversely also through increasing demand for forest expansion. This dynamic is highly complex and outside the scope of the current study. Here, we fully account for embodied emissions from harvesting and processing of imported HWPs. This approach is entirely appropriate for the 'consequential' LCA framing employed in this study, in which we account for all changes relative to 'BAU' for each modelled wood value chain scenario. This is explained in the Methods section, 'Scope of the LCA'.

We are able to assume that the emissions per product unit for each HWP are the same for UK-produced and imported products, on the basis that most sawmills use wood residues for heat energy and assuming similar average electricity generation emissions and sawmill efficiency across dominant source countries for UK softwood imports. We have added a sentence explaining this point to the Methods section, 'Process emissions' in the manuscript.

Finally, only the circular scenarios lead to a change (decrease) in virgin wood consumption and hence changes to harvest intensity (i.e. changes to terrestrial C relative to BAU). The GWP impact of this reduced demand for virgin wood is represented as reduced harvesting elsewhere (in the present study this is assumed to occur in imported-HWP source countries), albeit in a simplified, and probably conservative, way (using commercial forest similar to that of the UK as a proxy). This is explained in the Methods section, 'Terrestrial carbon'. NB. in these scenarios the volume of imported HWPs also decrease ('circular' by -15% and 'cascading&circular' by -5%).

12. The redirection of harvested wood in sawn wood and wood recycled medium density fiberboard (rMDF), do these products put any effect on the technical processing and socioeconomical changes in the respective sectors?

There are technical, investment, economic, employment etc. consequences of any change in the value chain, or adoption of any CE measure. In the present study, the responses of the interviewees were heavily influenced by their understanding of these issues. For example, they highlighted that better wood recovery, sorting and reporting is needed to facilitate identification and implementation of circular initiatives, for which a major finding of the study is the need for a mind-set change, as well as financial investment to implement. From this, the study identifies the need for policy change and advocacy to drive this change. rMDF technology is proven and ready for commercial scale-up, so we have added a line in the Introduction to make this clear. This presents an important contrast with future, as yet unproven technologies, as the interviewees raised the widespread challenges of conservatism in the value chain and that this both restricts prototyping and suppresses investment in new technology.

13. Authors also need to briefly discuss climate change parameters other than GWP. Also, is the data presented in normalized form? Authors are also suggested to study the variation in impact values if source of electricity is varied (from hard coal to natural gas).

We are not clear on what other GWP parameters the reviewer would like to be discussed, nor the meaning of 'normalised'. Results presented in the Figures are for aggregated GHG emissions (either annual or cumulative), expressed using GWP₁₀₀ and factors from IPCC AR5 – i.e. GWP₁₀₀ factors of 1, 28 and 265 for the main GHGs CO₂, CH₄ and N₂O. Using alternative approaches such as Global Temperature Potential or GWP* warming equivalent can be useful, especially when significant non-CO₂ forcing is involved (e.g. albedo effects, CH₄ – not the case in this study). Given that international GHG accounting and policy is based on GWP₁₀₀, and that the ranking of scenarios or main conclusions are not expected to be significantly impacted owing to the dominance of CO₂ forcing in our studied value chains, we think that exploring additional GHG aggregation metrics would add unnecessary complication to the interpretation of results. On the issue of varying electricity source, this is a good point but we already vary electricity generation emissions through time according to well-defined decarbonisation factors which are applied uniformly across all scenarios (coal electricity generation has already been effectively phased out in the UK).

14. The process of avoiding the emission by substituting the products is advocated to contribute the greatest GWP credit. The substitution of fossil fuel by wood fuel will still contribute to GWP impacts and might not be the best sustainable strategy for substitution of fuel sources. What would be the effect of substituting fossil fuels with renewable eco-friendly energy sources in this study? Similarly, substitution of concrete by sawn wood is also a less robust system comparatively. Utilization of forest biomass as commercial construction materials still has a long way to go and may require continuous maintenance.

We largely agree with the suggestions and underlying logic of the reviewer here, but respectfully suggest that they may have missed some of the methodological detail that addresses these issues directly in our study. Firstly, the whole study is predicated on the relative scarcity of wood, hence the need to utilise it more efficiently and in a targeted manner to ensure sustainability (we agree that wood cannot satisfy all energy and material demands!). The dynamic LCA employed in the study

already considers progressive decarbonisation of electricity and heat generation (and thus concrete emissions) through time, conferring decreasing substitution benefits on to wood. However, even optimistic decarbonisation projections involve some fossil fuel use beyond 2050 in difficult-to-abate sectors, such as cement production (UK Committee on Climate Change). Thus, changing the availability of wood to abate emissions in such sectors is likely to have GWP consequences for some time into the future.

The reviewer states that substitution of concrete by sawnwood is comparatively “less robust” but we are not clear on the meaning of this statement. However, we very much agree with the reviewer that there are many barriers to change in the construction sector’s use of different (renewable) materials and construction methods. We are pleased that the reviewer has raised this important point as it is a major understanding that underpinned the motivation for the second part of the study, and the responses of our interviewees. Nonetheless, we would emphasise the validity of considering additional use of wood in construction within our scenarios because the use of structural timber in building structures is well established in many parts of Europe and North America. Indeed, 75% of new houses built in Scotland are now timber framed, and there is high potential to increase the much lower proportion built this way across England, Wales and Northern Ireland. (Currently timber frame construction accounts for only 9% of new build homes in England, 22% in Wales, and 30% in Northern Ireland). Current UK policy already supports more timber-frame construction in place of concrete / brick, as a high-value (first) use of wood. Therefore, we think that inclusion of such use of wood in our scenarios is highly relevant.

15. Authors need to comment on the rationality for these avoided emissions by product substitution examples. At the same time, it is encouraged to propose more product substitution strategies.

We are unclear what is meant by ‘rationality’ in this point. If the reviewer is seeking further detail on our choice of product substitutions, we are confident that we do provide sufficient detail in the Methods section, ‘Substitution credits’. The product substitutions we used are already adopted in the UK and published in peer-reviewed literature (e.g. Forster et al. 2021). We purposefully selected them as the most realistic, appropriate examples, and the journal’s tight word limits for the paper prevent us from speculative discussion (without any new evidence) of other options that are less realistic within the context of our study system. Overall, we are confident that our approach is conservative with respect to our conclusions (e.g. we model rMDF with no direct non-wood substitution credits). But we do acknowledge that a wider range of substitution effects could arise in the future as material science and product development progresses – as we report in the additional supplementary information file, S10.

16. Implementation of cascading and circular wood materials flows in year 5 shows a higher rate of GWP impact decrease than implementation at year 10. However, both the implementation reaches the similar GWP value by the end of the time frame used for study period, i.e., 28 years. Which period of implementation would be better then? Please comment.

We assume that the reviewer is referring to Figure 2, which presents annual, not cumulative, GWP impact. The reason that the annual GWPs are the same (for scenarios with wood-use changes implemented in year 5 vs year 10) from year 10 onwards is because after this time both scenarios effectively become the same in their annual impact (as the same changes to the value chain have been implemented and the same decarbonisation factors are applied). However, the cumulative impact in Figure 3 shows noticeably different GWP impact results for the respective year 5 vs year 10

scenarios – which we discuss in Results subsection, ‘Early implementation optimises impact.’ We highlight the advantage of implementing circular and cascading initiatives in year 5 to take advantage of the increased cumulative impact that can be gained during years 5-10, which would be missed if implementation was delayed until year 10.

17. Decarbonization in BAU approach is implemented in stepped decadal decarbonization strategy. When compared with the cascading or circular approaches, the BAU with decarbonization only remains 5×10^5 CO₂ i.e., 25% of initial CO₂. I think instead of using cascading, circular, and both in combination, if only BAU with decarbonization is applied, then the current reform for diverting the wood flows, technical processing, and implementation of strategies would not be required. If the study period has been taken for more than 28 years, the BAU as well has reached to 0 CO₂ emission by the end of study period.

We thank the reviewer for raising this point. Yes, a continuation of the study period would achieve ‘net zero’ Scope 1-3 annual emissions for BAU (as we point out in the Results text), which is due to the future industrial decarbonisation assumptions used in the study. However, in observing the cumulative impact graphs (Figure 3), the cumulative emissions remain significantly higher and cannot be offset in a BAU scenario. It is important to note that the ‘circular&cascading’ scenario’s cumulative emissions (Figure 3) start to decrease at the end of the study period. The results show clearly that shifting from BAU to circularity leads to significantly lower cumulative emissions during the study period, which has important implications for both climate change mitigation and for national net zero GHG emissions targets – whereby sectors that are easier to decarbonise and sectors that offer carbon off-setting opportunities can/must make a greater contribution sooner.

Taking a passive approach to ‘decarbonising’ (i.e. relying on other sectors, e.g. the electricity generation and supply system, to decarbonise) would ignore the important impact the forestry value chain can achieve by reducing cumulative emissions in the interim, and by carbon offsetting (negative emissions) in the longer term.

We raise these points in both the Results and Conclusion sections:

“implementing ‘cascading and circular’ wood use could deliver 78% more cumulative climate-change mitigation by 2050 than decarbonisation of linear wood value chains alone.” (Updated excerpt from the Results section)

“only ‘circular’ or ‘cascading&circular’ wood use can lead to the forestry value-chain becoming an enduring (Scope 1-4) net carbon sink (even before considering the potential contributions of afforestation and bioenergy with carbon capture and storage (BECCS)).” (Excerpt from the Conclusion).

18. Pulp and paper industry is an important component of the forest value chains. Please comment on the rationality of its exclusion from the study.

The reviewer is correct in identifying the pulp and paper industry as an important component of forestry value chains. As explained in the Methods section, ‘Scope of the LCA’, we excluded it from the boundary of our study because in the UK the pulp and paper industry is essentially self-contained (and already achieves very high recycling rates). There is very little interaction between the pulp and paper industry and the rest of the wood value chain – therefore this component would remain unchanged across all the modelled scenarios. Also, due to its high associated process emissions it

would dominate the results, thus obscuring the consequential GWP impacts of the modelled circular and cascading scenarios (the focus of the study).

19. The resolution of all the figures needs to be improved and formatted (CO₂ and not CO2) as per the journal guidelines.

We thank the reviewer for pointing out that, while the 2 in CO₂ has the appropriate subscript in figure 1, we had failed to do this in figures 2 and 3 and have now corrected this. We have also increased the font size on the Figures to improve readability, as mentioned also in our response to Reviewer 1, point 4.

20. The authors need to check the grammatical errors, language, and formatting errors in the manuscript.

We thank the reviewer for this observation and can assure the reviewer that we have checked the manuscript carefully for any grammatical, language and formatting errors and have corrected any found.

21. Line 393: "Potential climate mitigation effect" what does it mean?

We have edited the text to read 'climate-change mitigation', to be consistent with terminology used throughout the paper. In this context, in line 393, it is synonymous with 'GWP impact'.

22. Authors need to format the referencing properly. The current form of referencing is very poor and not as per journal guidelines.

We thank the reviewer for drawing this to our attention. We have rechecked our referencing and have implemented a number of minor edits to make it even more closely matched to the journal guidelines.

Reviewer #2 (Remarks to the Author):

The manuscript entitled "Circular wood use can accelerate global decarbonisation but requires cross-sectoral coordination" identified the potential of carbon storage and greenhouse gas reduction in the UK wood-based value chain. This study covers cascading and circular wood use and afforestation as strategies to contribute in the Paris Agreement targets achievement based on a dynamic lifecycle assessment (LCA). For such, the authors presented barriers based on stakeholder interviews and proposed enablers to plan a and societal change system (SCS). To the best of my knowledge, this is a novel research that combines LCA and SCS to support a robust decision making in the wood sector, which can be applied in different contexts. In addition, the data are presented in the Supporting Information for transparency. Therefore, I recommend the publication of this manuscript.

We thank the reviewer for their endorsement of our paper.

Reviewer #3 (Remarks to the Author):

Dear authors

The manuscript provides a significant scientific contribution to devising future climate mitigation strategies for forestry-based sectors. However, here are a few comments that need to be addressed.

1. The term 'cascading scenario' is misleading. Cascading is the sequential use of wood in multiple applications. In your case, you have considered only 1 aspect of cascading: prioritising primary wood use for high-value applications (i.e. sawn wood). You refer to secondary wood use as 'circular'. I think the terminologies could be adapted to avoid confusion.

A relevant reference in this regard is - Navare, K., Muys, B., Vrancken, K. C. & Van Acker, K. Circular economy monitoring – How to make it apt for biological cycles? *Resour. Conserv. Recycl.* 170, 105563 (2021).

In response to the reviewers first concern, the reviewer correctly points out that hierarchical wood use is one (important) aspect of cascading use; with 'sequential use of wood in multiple applications' also important. While the reviewer correctly identifies that the major change from BAU in our enhanced cascading wood use example is enhanced hierarchical use, further cascading uses (i.e. 'sequential use of wood in multiple applications') are also modelled. In the UK there is very little landfilling of wood waste, therefore, our 'BAU' scenario already embodies significant elements of cascading wood use: recovered wood-use in particle board manufacturing (which has 50% recycled wood content in UK) and for bioenergy. We have added a sentence to make this clear, in Figure 5 in the Methods section. The enhanced cascading scenarios embody both a higher degree of hierarchical use and multiple sequential uses. By virtue of the improved hierarchical use, there are inherent increases in onward cascading use (sawnwood to particleboard, other wood to bioenergy – proportionate to the way recovered waste wood is used in the UK) and we capture this impact as increases in the HWP C pool (as described in Methods section, 'HWP carbon'). We respectfully maintain, therefore, that our use of the term 'cascading scenario' is not misleading.

We acknowledge that there are many other examples of cascading wood use that would benefit from further study (e.g. use of recovered sawnwood in cross-laminated timber). Indeed, this is a key finding of our study. However, as we set out in our response to reviewer 1, point 3, there is currently insufficient data (national level material flow of waste wood – quantity and quality) available to quantify the impacts of prospective alternative cascading scenarios through rigorous LCA. On the other hand, the improving hierarchical wood use in our scenarios is a realistic and quantifiable example of an important aspect of enhanced cascading use in the UK that is a foundational first step to optimising cascading use (i.e. sequential use in multiple applications). This is an important example of a scenario for which our findings for the UK case study also have direct relevance for other countries.

We thank the reviewer for alerting us to the Navare et al. (2021) paper and we are pleased to add a citation to this reference in the Results section of our paper, 'Change initiatives for progress'. It highlights the key features often missing from Circular Economy principles that uniquely apply to renewable resources, including cascading use and sustainable extraction. These are both important aspects that motivated the present paper – to generate quantitative evidence highlighting their importance. We believe these aspects should be additional to existing established CE principles, i.e. that circular wood use, where possible, is also important. An important finding of our study is that circular use can help to reduce (growing) demand for biological resources, and maintain sustainable extraction. Navare et al (2021) point out that in many cases circularity isn't possible in wood value

chains because of degradation of resources with each successive use, and this reinforces the importance of the kinds of enhanced cascading use scenarios included in our study. However, this issue of resource degradation is not inherently the case for circularity approaches in the wood value chain, and the recycling of MDF scenario tested in our study is an important example of where this is not a major constraint. It is a near closed-loop process, whereby recovered waste MDF is re-pulped and reformed back into MDF. There are also other examples of circularity approaches that are increasingly implemented and which can overcome the constraints of resource degradation, including modular buildings that are designed to be dismantled and reused, and pallets that can be repaired and reused. Therefore, our overview of this issue is that there will be some losses through the product lifecycle but this is also the case for “closed loops” based on abiotic resources.

We therefore maintain that recycling MDF is genuinely a circular use (within the technosphere) and not a cascading use, and therefore our use of the terminology is justified. In addition, we are sure that it is a valid exemplar of the range of circular use approaches that are increasingly being adopted but are as yet at too early a stage of implementation to provide sufficient data for rigorous LCA.

2. You emphasise the need for cross-sectoral integration of sustainability assessment. However, your scenarios are centred around wood use. Forestry is included only by considering 'avoided harvest' and increased afforestation. Both refer to the amount harvested but not to other forest management strategies (e.g. increasing forest diversity, prioritising plantation of certain types of wood etc.)

We acknowledge that there are many other important ways that forestry can contribute towards mitigation of the climate and biodiversity crises, including many alternative forest management options. Certainly, more circular and cascading use of wood could increase opportunities to adapt forest management in many different ways to address these issues. Therefore, methodologically in the present study, we use reduced harvest intensity as a robust (realistic, albeit simplified) proxy for forest management effects.

We already analysed a broader range of alternative forestry scenarios using dynamic LCA in our previous study (Forster et al. 2021). The present study was designed to be additional and complementary to that. We feel that analysis of further forestry examples in the present paper would be too much of a duplication of the objectives of our previous study. While there are certainly more important research questions to be addressed about a wider range of forest management options, and their interaction with the subsequent wood value chain, this involves significant measurement and modelling complexities that would require multiple additional studies, and could not be added to the present one without taking the paper well beyond the journal's word limits.

3. You have assumed that reducing the harvest increases the carbon stock in the forest. Is that always the case? It depends on the state of the forests - leaving the additional biomass in the forest may decrease the overall sequestration over time as emphasised in Nabuurs, G. J. et al. First signs of carbon sink saturation in European forest biomass. *Nat. Clim. Chang.* 3, 792–796 (2013). It is essential to account for this in your analysis (Of course, this is keeping aside the other ecosystem services).

We are pleased to acknowledge the importance of this point raised by the reviewer. The Nabuurs et al. (2013) paper refers to the whole European forest resource, which is dominated by forests of much greater age/maturity than those in our study, which are comparatively young plantations that are a long way (in age) from those that are approaching sink saturation. However, importantly, our model does include the decline in annual increment as trees grow past the standard rotation age and so

accounts for this important effect. We have added a sentence adding this detail in Methods section, 'Terrestrial carbon'.

4. [Line 410 - 414] Could you specify the wood imports for different scenarios? I understand that import amounts remain the same, however, the product that is imported is different in each scenario. Could you make it explicit in the figure or the text?

It is important that we correct the potential misunderstanding of our methods raised here by the reviewer. While the total volume (i.e. domestically produced plus imported) of each HWP type is kept the same in all the scenarios, as the domestic value chain changes from 'BAU' the import volumes change in each scenario to maintain the total balance. Tables calculating these volumes are included in S2, S3 (sheet 'GWP_impact', cells N94-101) & S4 (sheet 'GWP_impact', cells N96-103). We have added a sentence in the Methods section, 'Wood flow scenarios,' to clarify this. It is also important to note that the processing emissions associated with imported HWPs are assumed to be the same as for domestically manufactured products (as explained in detail in our response to reviewer 1, point 11) – only the transport emissions are different. We have added an explanation of this assumption in the Methods section, 'Process emissions'.

Reference:

Forster, E.J., Healey, J.R., Dymond, C. and Styles, D. Commercial afforestation can deliver effective climate change mitigation under multiple decarbonisation pathways. *Nature Communications* **12**: 3831 (2021) <https://www.nature.com/articles/s41467-021-24084>

REVIEWERS' COMMENTS

Reviewer #1 (Remarks to the Author):

I have checked the author's response to the comments. I think the authors have responded appropriately to the comments. On further reading the manuscript, it appears to be fine. The authors have justified the comments and one of the constraints is data availability with respect to the current work. Otherwise, this work now addresses the remaining issues that were there.

Reviewer #3 (Remarks to the Author):

The comments have been well incorporated. For the remarks which were not incorporated, the authors provided appropriate justification.

RESPONSE TO REVIEWER COMMENTS

We sincerely thank the Reviewers for the valuable time and thought they have committed to reviewing our paper. We are pleased that our careful consideration of their previous comments and the resulting modifications to our manuscript and supporting information meets their satisfaction.

REVIEWERS' COMMENTS

Reviewer #1 (Remarks to the Author):

1. I have checked the author's response to the comments. I think the authors have responded appropriately to the comments. On further reading the manuscript, it appears to be fine. The authors have justified the comments and one of the constraints is data availability with respect to the current work. Otherwise, this work now addresses the remaining issues that were there.

We thank you for your acceptance of our manuscript.

Reviewer #3 (Remarks to the Author):

1. The comments have been well incorporated. For the remarks which were not incorporated, the authors provided appropriate justification.

We thank you for your acceptance of our manuscript.